



**Molecular signatures and formation mechanisms of particulate matter**
**(PM) water-soluble chromophores from Karachi (Pakistan) over**
**South Asia**
Jiao Tang[1], Jun Li[1], Shizhen Zhao[1], Guangcai Zhong[1], Yangzhi Mo[1], Hongxing Jiang [2],
Bin Jiang[1], Yingjun Chen[2], Jianhui Tang[3], Chongguo Tian[3], Zheng Zong[4], Jabir Hussain
Syed[5], Jianzhong Song[1], Gan Zhang[1]
[1]State Key Laboratory of Organic Geochemistry and Guangdong province Key
Laboratory of Environmental Protection and Resources Utilization, Guangdong-Hong
Kong-Macao Joint Laboratory for Environmental Pollution and Control, Guangzhou
Institute of Geochemistry, Chinese Academy of Sciences, Guangzhou, 510640, China
[2]Shanghai Key Laboratory of Atmospheric Particle Pollution and Prevention (LAP[3]),
Department of Environmental Science and Engineering, Fudan University, Shanghai
200433, China
[3]Key Laboratory of Coastal Environmental Processes and Ecological Remediation,
Yantai Institute of Coastal Zone Research, Chinese Academy of Sciences, Yantai,
264003, China
[4]Department of Civil and Environmental Engineering, Hong Kong Polytechnic
University, Hong Kong, 999077, China
[5]Department of Meteorology, COMSATS University Islamabad (CUI), Islamabad,
45550, Pakistan
Corresponding Authors:
Jun Li: junli@gig.ac.cn; Gan Zhang: zhanggan@gig.ac.cn



**Abstract.**
Excitation-emission matrix (EEM) fluorescence spectroscopy has been widely used to
characterize chemical components of brown carbon (BrC), yet the molecular basics and
formation mechanisms of chromophores decomposed by parallel factor (PARAFAC)
analysis are not fully understood. Here, water-soluble organic carbon (WSOC) in
aerosols from Karachi, Pakistan, were characterized with EEM spectroscopy and
Fourier transform ion cyclotron resonance mass spectrometry (FT-ICR MS). Three
PARAFAC components were identified, including two humic-like (C1 and C2), and
one protein-like (C3) species. Among them, the C2 shows the longest emission maxima
(~494 nm), and correlates tightly with the mass absorption efficiency at 365 nm
($MAE_{365}$), the character of BrC. Molecular families associated with each of the three
components were determined by Spearman correlation analysis between FT-ICR MS
peaks and PARAFAC component intensities. The C1 and C2 components are associated
with nitrogen-enriched compounds, despite that C2 more with higher aromaticity,
higher N content, and highly oxygenated compounds. The formulas associated with C3
include fewer nitrogen-containing species, with a lower unsaturated degree and
oxidation state. A dominant oxidation pathway for the formation of C1 and C2
components was suggested, notwithstanding their different precursor types. A large
number of formulas associated with C2 were found to be located in the "potential BrC"
region, overlapped with BrC-associated formulas, and readily correlated tightly with
$MAE_{365}$. This suggests that the compounds illuminating C2 may have also contributed
substantially to the BrC light absorption. These findings were important for future
studies using the EEM-PARAFAC method to explore the compositions, processes, and
sources of atmospheric BrC.

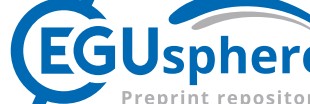

# 1. Introduction

Water-soluble organic carbon (WSOC), containing aromatic rings or aliphatic structures with carboxyl, hydroxyl, carbonyl, or methoxy functional groups, constitutes a significant portion of organic aerosols (OA) and can affect the roles of aerosols in climate processes, air quality, and human health (Lin and Yu, 2011; Wang et al., 2020b). Atmospheric WSOC originated from primary emissions, including biomass burning, coal burning, and other primary sources (Park and Yu, 2016; Li et al., 2018; Tang et al., 2020); as well as the secondary formations, such as the aqueous-phase reaction from anthropogenic or biogenic emission (Gilardoni et al., 2016; Lamkaddam et al., 2021; Updyke et al., 2012; Yu et al., 2021). Light-absorbing WSOC is an important component of brown carbon (BrC) and has a strong wavelength-dependent absorption that peaks in the ultraviolet (UV) spectral region and declines through the visible spectral region (Choudhary et al., 2022; Hecobian et al., 2010; Laskin et al., 2015; Sullivan et al., 2022). This fraction can contribute significantly to the global radiation balance and affects the photochemistry of the atmosphere (Feng et al., 2013; Kirchstetter and Thatcher, 2012).

The reactivity and fate of WSOC are tightly linked to its chemical composition, yet isolating and characterizing the constitutive elements and structure of light-absorbing organic molecules from among an abundance of non-absorbing molecules in aerosols is a challenging task. Bulk optical measurements, such as UV−visible (UV−vis) absorption spectroscopy, is an efficient means of characterizing the light absorption properties of BrC (Wu et al., 2021; Hecobian et al., 2010). Excitation-emission matrix (EEM) fluorescence spectroscopy can be employed to quantify and characterize a subset of BrC chromophores that absorb certain wavelengths of light and re-emit a fraction of that energy as fluorescence, providing further insight into the origin, chemical property, and process information. EEM is often coupled with parallel factor (PARAFAC) analysis, which mathematically decomposes the fluorescence data into



various components, such as humic-like (HULIS), protein-like components (PLOM),
and so on (Chen et al., 2016b; Jiang et al., 2022c; Tang et al., 2020; Chen et al., 2021a).
Due to its high detection sensitivity and chromophore-resolving ability, EEM
spectroscopy has been widely used to track BrC chromophore variations in chamber
experiments (Bianco et al., 2014; Lee et al., 2013; Vione et al., 2019), and in ambient
BrC studies (Wu et al., 2019; Yue et al., 2019; Chen et al., 2021b). Despite widespread
use, the molecular basics of atmospheric fluorescent chromophores are not fully
understood. Ultrahigh-resolution Fourier transform ion cyclotron resonance mass
spectrometry (FT-ICR MS) is a powerful tool facilitating the analysis of organic matter
based on individual molecular formulas that can aid in the interpretation of molecular
patterns across systems (Jiang et al., 2014; Mopper et al., 2007). Based on this technique,
thousands of molecular formulas can be obtained, and basic structural features can be
deduced (Song et al., 2019; Zeng et al., 2021).
The water-soluble fraction of total OC aerosol is commonly up to 70% (Li et al.,
2020b; Wu et al., 2019; Mo et al., 2022). Although serval studies have combined
fluorescence analysis and FT-ICR MS to characterize the atmospheric WSOC, few have
focused on the molecular characteristics of fluorescent components (Su et al., 2021;
Tang et al., 2020; Li et al., 2022). Hence, it is necessary to seek the molecular basis of
water-soluble fluorescent components.
Due to the chemical complexity of fluorescent chromophores, the formation of
these compounds may be complex and would be determined by the precursors and
atmospheric conditions (OH radicals, $O_3$, $NO_x$, and so on). For example, Lee et al. (2013;
2014) demonstrated that ammonia vapor converts initially colorless *d*-limonene/$O_3$
secondary organic aerosol (SOA) into BrC material, yet the *d*-limonene/$O_3$+$NH_3$ brown
material almost completely lost its ability to absorb visible radiation or fluoresce after
irradiation. Conversely, Bianco et al. (2014) examined that tryptophan, tyrosine, and 4-
phenoxyphenol in an aqueous solution after irradiation produced species with similar
fluorescence properties as humic substances. Fan et al. (2020) found that PLOM



components decomposed by PARAFAC analysis are quenching with $O_3$ aging, yet
HULIS components exhibit a gradual increase with $O_3$ aging. Recently, results based
on observations in the online EEM system showed a conversion process between
highly-oxygenated and less-oxygenated HULIS under atmospheric oxidation (Chen et
al., 2021a). However, studies on the molecular evidence were limited for the formation
of fluorophore compounds, especially PARAFAC fluorescent components; it is also
important to constrain the properties of BrC aerosols.
Here, we examined the FT-ICR MS and fluorescence spectroscopy for WSOC in
aerosols from Karachi, Pakistan in South Asia. Karachi represents a typical urban
setting with a variety of air pollution sources including industrial and vehicular
emissions, dust storms, and sea salts (Khwaja et al., 2009). The objectives of this study
are to obtain: (1) the molecular families associated with PARAFAC components by
using Spearman analysis of FT-ICR MS peaks and PARAFAC component intensities;
(2) the possible formation pathway of PARAFAC components based on their molecular
formulas; and (3) the molecular-level correlation between BrC chromophores and
PARAFAC components. The results obtained help to understand the composition and
fate of PARAFAC components, which broadens the application of the EEM-PARAFAC
method to characterize atmospheric BrC.
**2. Materials and Methods**
**2.1. Sampling campaign**
Total suspended particulate (TSP) samples were collected in Karachi (24°51′N;
67°02′ E) from 2 February 2016 to 27 January 2017, as described previously (Zong et
al., 2020). Briefly, the sampling site is on the floor of an office building of a government
agency (approximately 12 m in height) located at the west edge of the main district of
Karachi (Fig. S1 in Supplement). No agricultural land, but some semiarid bushes,
surround it. Samples were collected on glass fiber filters with a high-volume sampler
(KC-1000, Longtuo) at a flow rate of 1.13 $m^3$ $min^{-1}$. Diurnal sampling (12/12 h; day:
10:00 am−10:00 pm; night: 10:00 pm−10:00 am, local time) was carried out once a
week. In total, 96 samples were collected during the sampling campaign. All filters were
preheated at 450 °C for 6 h in a muffle furnace before sampling. After sampling, the
filters were folded and stored in a refrigerator (−20 °C) until further analysis. The
classification of seasons based on the air-mass trajectories contains pre-monsoon (Mar–
May), monsoon (Jun–Sep), post-monsoon (Oct–Nov), and winter (Dec–Feb), where
marine air masses were prevailing in the monsoon period and continental air masses
were popularly occurred in the other periods, as shown in Fig. 1. The meteorological
parameters over Karachi during sampling periods were provided in Table S1.

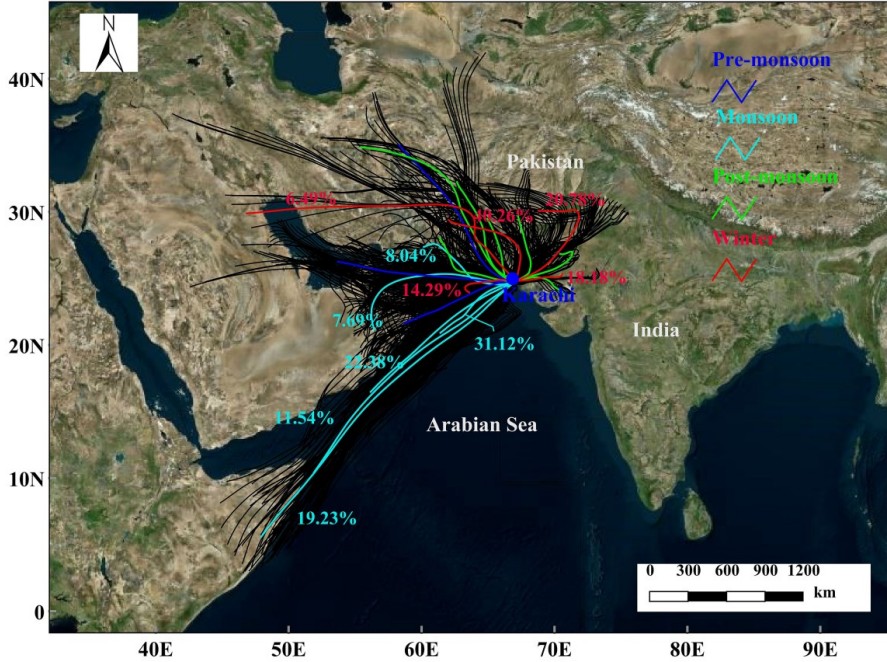


**Figure 1**. The 72 h back air-mass trajectories at Karachi (Pakistan) from February 2, 2016, to
January 27, 2017. The samples were classified into four seasons, whose air masses were further
conducted by the cluster analysis, including pre-monsoon (blue lines), monsoon (azure lines), post-
monsoon (green lines), and winter (red lines). The air-mass trajectories were analyzed by the
HYSPLIT model. The base map was derived from Bing Maps (© 2023 Microsoft).





**2.2. Optical characterization and PRAFAC modeling.**
The detailed methods of fluorescence characterization and PARAFAC analysis
have been described in our previous studies (Tang et al., 2021; Tang et al., 2020). Briefly,
WSOC was obtained by ultrasonicating the filter punches in ultrapure deionized water
(resistivity of > 18.2 MΩ) for 30 min and filtering the solutions. UV–vis absorption
spectra and EEM spectra were simultaneously collected using a fluorometer (Aqualog;
Horiba Scientific, USA). UV–vis absorption spectra were scanned in the range of 239
to 800 nm with a step size of 3 nm. Fluorescence scans were collected over increments
of 3 and 4.66 nm for the excitation (239–800 nm) and emission (247–825 nm)
wavelengths, respectively. The corresponding concentration of WSOC and water-
soluble total nitrogen (WSTN) was quantified using a TOC analyzer (Vario TOC cube;
Elementar) (Yu et al., 2017). The quality control for absorption spectra and EEM spectra,
as well as carbon mass, were detailed in Text S1 of the supplement.
PARAFAC analysis was conducted to decompose the EEM datasets, using the
drEEM Toolbox in MATLAB version R2016a (http://models.life.ku.dk/drEEM, last
access: June 2014) (Murphy et al., 2013) (details in Text S2). The light absorption
coefficient (Abs$_\lambda$, Mm$^{-1}$), mass absorption efficiency (MAE, m$^2$ g$^{-1}$ C), absorption
Ångström exponent (AAE), and optical indices, such as specific ultraviolet absorbance
(SUVA), spectral slope (S$_R$), as well as fluorescence-based indices of the fluorescence
index (FI), biological index (BIX), and humification index (HIX), were presented in
Text S2.
**2.3. FT-ICR-MS analysis.**
A representative subset (12 samples) that covered a range of values of the relative
intensities of PARAFAC components, WSOC concentrations, and total organic nitrogen
at day and night in different seasons, was selected for ultrahigh-resolution electrospray
FT-ICR MS analysis. Specially, the 72 h back air-mass trajectories of the selected
samples showed different sources of air pollution (Fig. S2). Hence, the selected samples



reflected the typical features during the sampling period. The sample preparation and
analysis of FT-ICR MS are given elsewhere (Jiang et al., 2021; Tang et al., 2020).
Briefly, the WSOC extracts were desalted and concentrated through the solid-phase
extraction (SPE) method (Xu et al., 2020; Zhou et al., 2021), using a hydrophilic-
lipophilic balance (HLB) cartridge (Oasis HLB, 200 mg/cartridge, Waters, USA) (Varga
et al., 2001). The efficiency of the SPE method was evaluated by measuring the carbon
masses, UV–vis, and EEM spectra before and after elution (Varga et al., 2001), which
showed good analytical recoveries (details in Text S3). The majority of inorganic ions,
low-molecular-weight (MW) organic acids, and sugars were not retained by the SPE,
and the constituents retained on the SPE cartridge were eluted with methanol containing
2 % ammonia (v/v) (HPLC grade) (Lin et al., 2010). Methanol containing 2 % ammonia
was selected for elution to reduce the mass percentage of irreversibly adsorbed carbon
but not significantly change the molecular composition (Chen et al., 2016a; Lin et al.,
2012a). The eluents were dried under a gentle nitrogen gas stream and redissolved in 1
mL of methanol for subsequent analysis.

Preceding FT-ICR-MS analysis, ultrahigh-resolution mass spectra were obtained

using a solariX XR FT-ICR MS instrument (Bruker Daltonics GmbH, Bremen,
Germany) equipped with a 9.4 T superconducting magnet and an electrospray
ionization (ESI) ion source. Ions were produced in negative and positive ESI ion mode
(hereinafter abbreviated as ESI– and ESI+). The ion accumulation time was set to 0.6,
and the $m/z$ range was set to 150–800. Peaks were considered if the signal-to-noise ratio
was greater than 4. A typical mass-resolving power of > 450 000 at m/z 319 with < 0.2
ppm absolute mass error was achieved. The identified formulas were classified into
subgroups (CHO–, CHON–, CHOS–, CHONS–, CHO+, CHON+, CHONa+, and
CHN+). The CHO– and CHO+ refer to those that contain carbon, hydrogen, and oxygen
and are detected in the ESI– and ESI+ modes, respectively. Other compound categories
are defined analogously. Note that we report all detected compounds as neutral species
unless stated otherwise. The double bond equivalent (DBE), modified aromaticity index



($AI_{mod}$), carbon oxidation state ($\overline{OS}_C$) and the nominal oxidation state of carbon (NOSC)
are calculated in Text S4. The formulas were further classified into four categories
referring to the $AI_{mod}$ and H/C ratio (Kellerman et al., 2015; She et al., 2021): condensed
aromatic compounds ($AI_{mod} \geq 0.67$); aromatic compounds ($0.5 < AI_{mod} < 0.67$); highly
unsaturated and phenolic compounds (H/C $\leq 1.5$, $AI_{mod} \leq 0.5$); and aliphatic compounds
($1.5 \leq$ H/C $\leq 2.5$).

### 2.4. Statistical analyses

The molecular families associated with each PARAFAC component were derived

using Spearman correlation analysis according to our previous method (Jiang et al.,
2022c). Before analysis, the PARAFAC component intensities were normalized to the
total fluorescence intensity for a given sample, and the intensities of FT-ICR MS peaks
were normalized to the total intensity of all peaks to which formulas were assigned
within a sample. Then, Spearman's correlations between the FT-ICR MS and
PARAFAC component data were conducted in R. Note that FT-ICR MS peaks that were
present in fewer than two samples were not considered. For an n of 12 samples, a
Spearman's coefficient ($r_s$) of 0.57 was calculated to be significant at the 95%
confidence limit (Student's t test, see Text S5). When Spearman's $r_s \geq 0.57$, the
molecule was assigned to the PARAFAC components. The same method was used to
obtain the molecular signatures of PARAFAC components of dissolved organic matter
(DOM) in aquatic environments (Stubbins et al., 2014; Singer et al., 2012). The
molecular formulas associated with Abs at 365 nm ($Abs_{365}$), and optical indices ($S_R$,
$SUVA_{254}$, $A_{254}$, FI, BIX, and HIX) were similarly obtained. Note that many of these
associations may be due to different compounds responding in the same way to
environmental conditions.





## 3. Results and Discussion

### 3.1. Fluorescence and light absorption properties

PARAFAC analysis identified three individual fluorescent components (Fig. 2a), two humic-like (C1, C2), and one protein-like (C3) (Ishii and Boyer, 2012; Coble, 2007; Wu et al., 2019). All of them have been ubiquitously detected in aerosols (Wu et al., 2019; Wen et al., 2021; Wang et al., 2020a; Han et al., 2020), snow (Zhou et al., 2021), and rainwater (Li et al., 2020c). Based on the comparison with the EEM region of fluorescent components classified by a previous study (Chen et al., 2016b), C1, C2, and C3 were similar to less-oxygenated and highly-oxygenated HULIS, and non-N-containing components, respectively. Note that the origins and chemical properties of the components defined in this study are not necessarily similar to those of components with the same name in other organic matter.

The relative abundances of PARAFAC component intensities were used to indicate the changes in chemical compositions. Humic-like components (C1 and C2) were the dominant species of the whole-year samples, representing 80 %±6.1 % of total fluorescence intensity (Fig. S3), which is in accordance with precipitation samples in the Guanzhong basin of China while higher than that at Seoul (Yan and Kim, 2017; Li et al., 2022). Of those, C1 is 2-fold greater than that C2. During the sampling period, measurements of water extracts revealed the differences in PARAFAC component distributions in different seasons. When the monsoon is prevailing (marine-derived air masses, Fig. 1), the contribution of humic-like components decreased significantly (t-test, $p < 0.01$), suggesting that humic-like components may have a terrestrial origin. Recent work revealed that continental-influenced WSOC was enriched in aromatic and higher oxidation level compounds, while saturated primary marine biological compounds with lower oxidation levels were more abundant in marine-influenced WSOC (Mo et al., 2022). The redundancy analysis (RDA, detail in Text S6) conducted on PARAFAC components and noncollinear variables (variance inflation factors, VIF





< 10) showed that C2 was positively related to HIX, $A_{254}$, and $SUVA_{254}$, and C1 was
positively associated with HIX, yet C3 correlated with FI, BIX, and $S_R$ (Fig. S4). The
results reflected that C1 and C2 are associated with a high degree of humification and
aromaticity (Zsolnay et al., 1999; Weishaar et al., 2003).

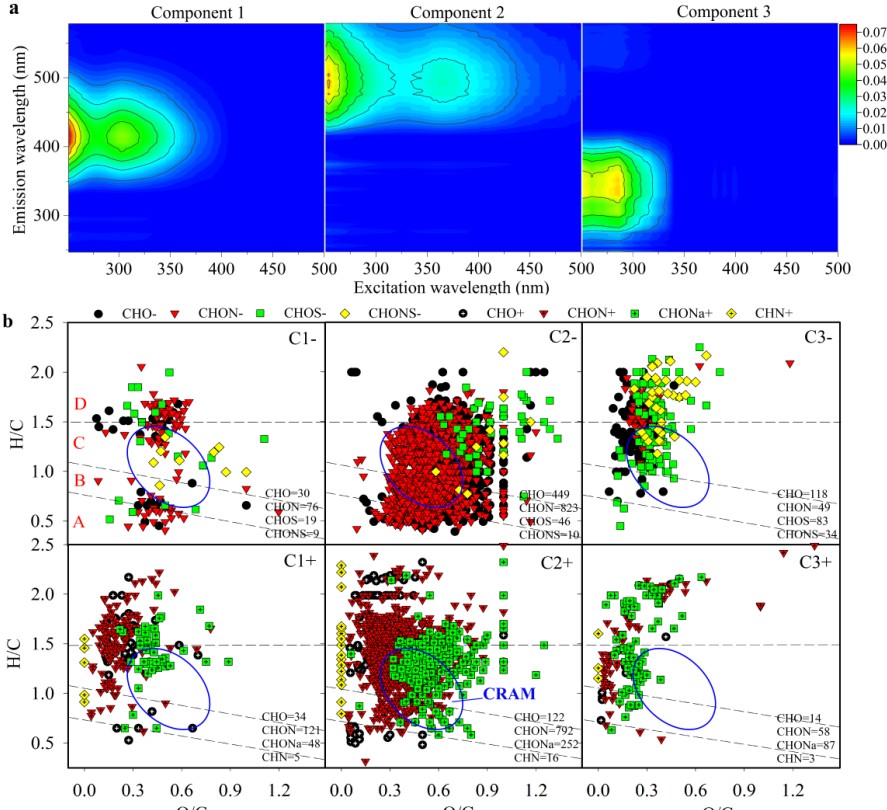

**Figure 2**. Molecular characteristics of PARAFAC components. (**a**) Patterns of the fluorescent
components (C1-3) identified by using the PARAFAC method. (**b**) van Krevelen diagrams of FT-
ICR MS-identified compounds assigned to PARAFAC components (C1, C2, and C3). C1- and C1+
refer to the molecules that are assigned to C1 detected in the ESI– and ESI+ modes, respectively,
and the others are defined analogously. Carboxylic-rich alicyclic molecules (CRAM) are remarked
by the blue ellipse (Hertkorn et al., 2006). The dotted line shows the class identification, including
condensed aromatic compounds (A); aromatic compounds (B); highly unsaturated and phenolic
compounds (C); and aliphatic compounds (D).

The mass absorption efficiency at a wavelength of 365 nm ($MAE_{365}$) is a key

parameter that can be used to describe the light-absorbing ability of the different





chromophores in BrC. The $MAE_{365}$ of water extracts in Karachi were 0.80±0.40 m$^2$ g$^{-1}$
C, which is comparable to that in Kharagpur (India) (Srinivas and Sarin, 2014) and
Godavari (Nepal) (Wu et al., 2019), higher than in Indo-Gangetic Plain (IGP) outflow,
such as Northern region of Maldives and Bay of Bengal (Bosch et al., 2014; Srinivas
and Sarin, 2013), yet much lower than other cities in South Asia (Table S2, and Fig. 3c).
The values of $MAE_{365}$ in monsoon daytime (nighttime) and non-monsoon daytime
(nighttime) are 0.47±0.20 m$^2$ g$^{-1}$ C (0.71±0.30 m$^2$ g$^{-1}$ C) and 0.88±0.34 m$^2$ g$^{-1}$ C
(0.92±0.48 m$^2$ g$^{-1}$ C), respectively (Fig. 3a). The $MAE_{365}$ exhibited strong seasonal
variation and was much higher at non-monsoon than in the monsoon (test-t, $p < 0.01$),
potentially due to the different sources and formation process. However, considerably
increasing $MAE_{365}$ values during nighttime than daytime in monsoon were observed (t-
test, $p < 0.05$), perhaps indicating that more light-absorbing substances were formed
during nighttime $NO_3^-$ reaction and significantly enhanced the light absorption capacity
of BrC in monsoon (Li et al., 2020a). No obvious increase was observed in non-
monsoon season (t-test, $p > 0.05$).

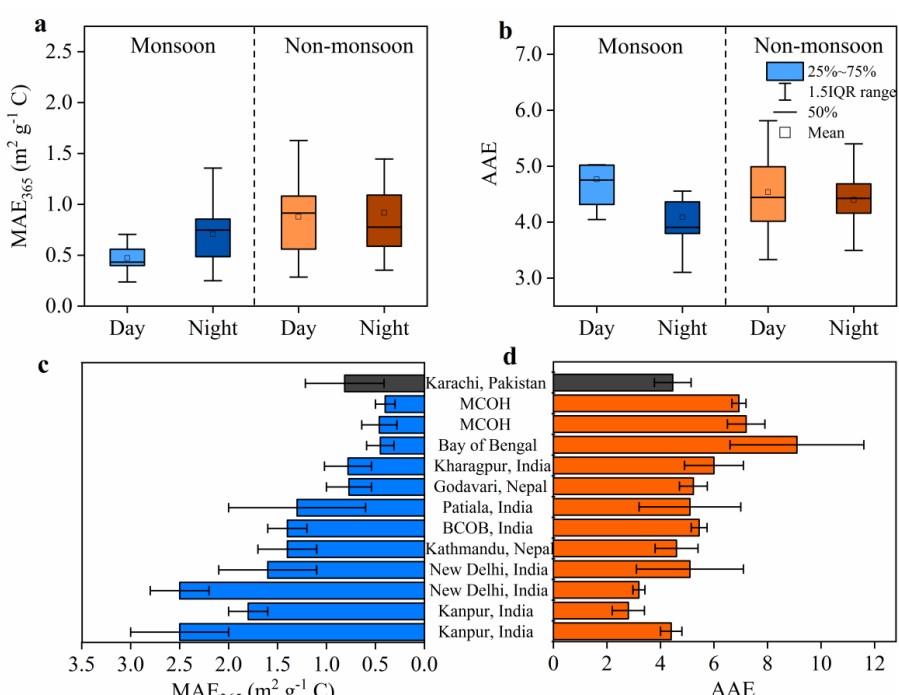

**Figure 3**. The seasonal and diurnal distributions of MAE$_{365}$ (**a**) and AAE (**b**) of WSOC in Karachi, Pakistan. Panels **c** and **d** refer to the MAE$_{365}$ and AAE of WSOC reported in recent studies over South Asia, including the Northern region of Maldives (MCOH) (Dasari et al., 2019; Bosch et al., 2014), Bay of Bengal (Srinivas and Sarin, 2013), Kharagpur (Srinivas and Sarin, 2014), Godavari (Wu et al., 2019), Patiala (Srinivas et al., 2016), Bhola Island in the delta of Bay of Bengal (BCOB) (Dasari et al., 2019), Kathmandu (Chen et al., 2020), New Delhi (Kirillova et al., 2014; Dasari et al., 2019), and Kanpur (Choudhary et al., 2018; Choudhary et al., 2021). The column marked in black color represents our results.

The parameter AAE reflects both the wavelength dependence of light absorption and the conjugated degree of the extracted compounds. The average AAE values (fitting at 330–400 nm) of WSOC in Karachi were very close in monsoon (4.4±0.85) and non-monsoon season (4.5±0.62) (Fig. 3b). However, the AAE values of WSOC during monsoon nighttime substantially decreased (t-test, $p < 0.05$), indicating a more aromatic and conjugated level of WSOC during nighttime, consistent with the significantly increased MAE values in this season. Except for in the IGP outflow, the AAE values of WSOC in Karachi were similar to that in South Asia, only higher than Kanpur and New Delhi (Fig. 3d). In addition, as humic-like components and MAE$_{365}$ of WSOC





significantly decreased in monsoon season and increased in non-monsoon season, the
correlation between them was carried out (Fig. S5). The result showed that only C2
positively correlated with $MAE_{365}$ (r = 0.49, $p$ < 0.01), suggesting that C2, representing
a category of compounds, potentially makes significant contributions to the BrC light
absorption. To ascertain this preliminary conclusion, we further investigated the
molecular composition of fluorescent chromophores, as discussed below.

### 3.2. Molecular signatures of PARAFAC components

Using Spearman's rank correlations at the 95 % confidence limit, 22 % of total
ESI– formulas and 23 % of ESI+ formulas were assigned to the three PARAFAC
components, respectively. The elemental formulas of each PARAFAC component are
listed in Table S3, and the elemental composition characteristics of complex molecular
formulas of PARAFAC components are presented by the Van Krevelen (VK) diagrams,
as shown in Fig. 2b.
The C1 was assigned 134 ESI– formulas and 208 ESI+ formulas, in which only
one formula overlapped in both modes. C1-assigned formulas were enriched in N-
containing molecules, accounting for 63 % and 61 % of its assigned ESI– and ESI+
formulas, respectively. Of those, CHON was the most prevalent. Given that the C1-
assigned formulas contain abundant nitrogen and are oxygenated ($\overline{OS}_C$, and NOSC)
(Kroll et al., 2011), $NO_x$-derived reactivity could occur in the formation of this
component (He et al., 2022; Lee et al., 2014). In contrast to C2 and C3, C1 formulas
had middle elemental composition ratios (e.g., O/C, H/C, and N/C), oxidation state
($\overline{OS}_C$, and NOSC), as well as unsaturated degrees ($AI_{mod}$, and DBE) and relatively
higher MW (Table S3, Figs. S6–S7). Of the formulas assigned to C1, 34 % of ESI–
formulas and 50 % of ESI+ formulas were aliphatic compounds, 35 % and 39 % were
highly unsaturated and phenolic compounds, and 31 % and 11 % were aromatic
compounds.



A larger number of 1328 ESI– formulas and 1182 ESI+ formulas were assigned to
C2, of which 108 formulas were overlapped. In common with C1, 67 % of ESI–
formulas and 68 % of ESI+ formulas contained nitrogen, suggesting that C2 is also an
N-enriched component. However, C2 formulas contain two or more nitrogen compared
with C1 (N atom $\geq$ 2, 38 % vs 14 % in ESI– and 42 % vs 18 % in ESI+) (Table S4),
perhaps because they have different precursors or origins. For example, $CHON_2$ species
are observed almost exclusively in the aged limonene ozonolysis sample than in $a$-
pinene ozonolysis (Laskin et al., 2014). C2 with the longest emission maxima (~ 494
nm), was assigned to more compounds in higher aromaticity ($AI_{mod}$ > 0.5), higher
oxidation state ($\overline{OS}_C$ > 0), and higher N content (N/C > 0.2) (Fig. S6). Accordingly, the
aromaticity index, N/C ratio, and NOSC correlated maxima with fluorescence in a
pattern strikingly similar to C2 (Fig. S8), consistent with a previous study (Kellerman
et al., 2015). Specially, C2 formulas contain more compounds in the region of
carboxylic-rich alicyclic molecules (CRAM) that are represented by carboxylated and
fused alicyclic rings with very few hydrogen atoms in double bonds, yet not common
for C1 and C3, suggesting that C2 has a probable character of resistance to
biodegradation and refractory nature (Hertkorn et al., 2006). The findings collectively
indicated that C2 is a highly aromatic, oxygen-rich, and high N-content component,
with potentially recalcitrant properties in the atmosphere.
Lower than 4 % ESI– and ESI+ formulas were assigned to C3. In contrast, more
than 60 % of formulas contain no nitrogen atoms (Table S3), implying that the
atmospheric protein-like component is not exclusively associated with N-containing
compounds, consistent with previously reported (Stubbins et al., 2014). A possible
explanation is that lignin, simple phenols, and naphthalene have a strong fluorescence
signal in this region (Hernes et al., 2009; Maie et al., 2007; Wu et al., 2019). Of the
formulas assigned to C3, almost all aromatic ESI– compounds and 43 % of aromatic
ESI+ compounds are without nitrogen, suggesting that a portion of water-soluble
protein-like components were substantially derived from N-free or N-depleted aromatic



compounds, which corroborated the previous results (Chen et al., 2016b). However, an
increasing fraction of S-containing species was observed, representing 41 % of C3-
assigned ESI– formulas. Although these S-containing compounds may have no
substantial contribution to BrC chromophores or fluorophores, similar formation
pathways or origins is the potential to reflect C3 fate. C3 was strongly associated with
less aromaticity degree ($AI_{mod} < 0.5$) and a higher degree of saturation compounds (H/C >
1) (Fig. S6), hence, lower than 5 % of formulas assigned to C3 were aromatic and
condensed aromatic compounds. This character coupled with less oxidation state ($\overline{OS_C}$
< 0, O/C < 0.5) collectively indicate that the atmospheric protein-like component is low
conjugation, oxygen-depleted, and N-depleted species.

The use of Spearman's correlations purposefully allowed molecular formulas to

correlate with one or more PARAFAC components as a given FT-ICR MS molecular
formula can include many different structural isomers. Thus, the common molecules
assigned to different PARAFAC components could preferentially reflect their similar
chemical structures, molecular compositions, and origins. Of the formulas assigned to
PARAFAC components, C1 shares 52 ESI– formulas and 42 ESI+ formulas with C2
(Fig. S9). The overlapped formulas represented 39% of 134 ESI– formulas and 20% of
208 ESI+ formulas assigned to C1. This observation indicates that C1 possesses a
molecular-level correlation with C2, suggesting that they may have similar origins and
atmospheric processes, as discussed below. However, no common molecules were
found between C3 and C1 and C2, respectively.
**3.3. Potential formation mechanisms of PARAFAC components**

Early studies suggested that C1 and C2 components are less oxygenated and highly

oxygenated humic-like components, respectively (Chen et al., 2016b), and another
study found the two components may transform each other under atmospheric oxidation
(Chen et al., 2021a). However, the formation pathways for these components are not
fully understood. Conversely, fluorescence changes along with molecular composition



were observed in the photolysis experiment of naphthalene-derived SOA (Lee et al.,
2014). HULIS-type fluorescence was also found to be produced in the vanillin and
acetosyringone solutions under simulated sunlight, attributed to oligomerization
processes observed from their molecular composition (Vione et al., 2019). These results
indicated that the molecular signatures could reflect possible formation pathways of
fluorescent components. It is worth noting that the selected samples for FT-ICR MS
analysis were collected on the day and night in different seasons, yielding a relatively
complete molecular dataset, thus to better explain the formation pathways of
fluorescent component-associated formulas we considered the potential precursors
from the molecular dataset. The reason is that the precursors may be not associated with
fluorescent components not as same as their formed new compounds, resulting in an
underestimation of the contribution.

The molecular composition of PARAFAC components was determined, including

CHO–, CHON–, CHOS–, CHONS–, CHO+, CHON+, CHONa+, and CHN+
compounds. For the different groups, the pathways may substantially differ. Given that
the N-containing compounds were enriched, in particular for C1 and C2, the formation
mechanism of this group was discussed first. N-containing compounds contain CHON–
(CHONS– was classified as S-containing compounds), CHON+, and CHN+ (few were
associated with PARAFAC components and no longer discuss). Of the CHON formulas
assigned to PARAFAC components, the organic nitrogen molecules are suggested to be
divided into the subgroups by using the O/N ratios, such as oxidized forms with O/N >
2 ($-NO_2$ or $-ONO_2$), reduced forms with O/N < 2 (amino or amide groups), which has
been widely used for the classification of CHON molecules in FT-ICR MS studies (Mo
et al., 2022; Jiang et al., 2022b; Zeng et al., 2021). A high relative abundance of CHON
subgroups in aerosol samples appeared O/N ≥ 3, generally like to contain numerous
oxidized nitrogen function groups (Mo et al., 2018). In this study, 87 % of CHON– and
74 % of CHON+ molecules assigned to C1, 84 % and 63 % of formulas assigned to C2,
and 86 % and 33 % of formulas assigned to C3, had O/N ≥ 3 (Figs. 4a, 4b, 4c and S10a,





S10b, S10c). Of those, the oxidized CHON assigned to C1 and C2 may be underestimated due to their formulas containing two-, and two more N atoms and need no 3 folds of O atoms to form $-NO_2$ or $-ONO_2$ groups. Hence, we assumed that these CHON formulas assigned to PARAFAC components were largely in an oxidized form.

Recent work summarized the several pathways for organic nitrates, including oxidation-product pair, hydrolyzation-product pair, mixed-processes product, and the remaining unknown product (Su et al., 2021). Kames et al. (1993) found that alcohols, diols, and hydrooxyketones reacting with $N_2O_5$ could produce organic nitrates: $R_1OH + N_2O_5 \rightarrow R_1ONO_2 + HNO_3$. We thus defined $R_1OH$ and $R_1ONO_2$ as an oxidation-product pair, with an element difference of $-H+NO_2$. In addition, the organic nitrates are suggested to undergo hydrolysis, with the formation of $HNO_3$: $R_2ONO_2 + H_2O \rightarrow R_2OH + HNO_3$; $R_2ONO_2$ and $R_2OH$ are defined as a hydrolyzation-product pair. Sometimes, $CHON_1$ can define as an oxidation-product pair with CHO and a hydrolyzation-product pair with $CHON_2$. If the two processes occur simultaneously, we defined this molecule as a potential mixed-processes product. A more detailed description of the three processes was presented elsewhere (Su et al., 2021).

Figures 4d, and 4e show potential pathways of CHON− molecules (CHON+ in Fig. S10) assigned to C1, C2, and C3, respectively. The result showed that oxidation-product pair, hydrolyzation-product pair, and mixed-processesproduct could explain a significant proportion, with 33 %–75 %, 69 %–77 %, and 44 %–80 % of CHON formulas assigned to C1, C2, and C3, respectively, which is comparable to the explained proportion of 52.8 %–69.7 % for $CHON_1$ and 43.4 %–53.5 % for $CHON_2$ in snow samples (Su et al., 2021). $CHON_1$ formulas assigned to C1 and C2 exhibited the highest proportion of oxidation from CHO, yet the primary precursor types are aliphatic compounds (52 %) and highly unsaturated and phenolic compounds (30 %) for C1 and highly unsaturated and phenolic compounds (54 %) and aromatic/condensed aromatic compounds (40 %) for C2, respectively (Figs. 4f, and 4g). Whereas $CHON_1$ assigned to C3 exhibited the highest proportion of mixed-processes product. Note that C1 and





C3 formulas contain fewer $CHON_2$ and $CHON_3$ formulas, which is no longer to discuss
them. Conversely, C2 formulas contain abundant $CHON_2$ and $CHON_3$ formulas, and
the potential oxidation-product pairs from $CHON_1$ ($CHON_2$) contributed to 69 % of
$CHON_2$ (77 % of $CHON_3$), largely deriving from highly unsaturated and phenolic
compounds and aromatic compounds. In contrast to ESI–, the contribution determined
by the three processes decreased with the portion of 5 %–73 %, 29 %–65 %, and 0–17 %
for C1–3-assigned CHON+ formulas, respectively. Due to the small number of $CHON_{2-3}$
formulas assigned to C1 and $CHON_{1-3}$ formulas assigned to C3, we only discussed
$CHON_1$ of C1 and $CHON_{1-3}$ of C2 (Fig. S10). The hydrolyzation process contributed
38 %, and oxidation-product pairs contributed 30 % of C1-assigned $CHON_1$. In contrast,
oxidation-product pairs contributed to 29 %–48 % for C2-assigned $CHON_{1-3}$ formulas,
yet the main precursor types are highly unsaturated and phenolic (56 %–70 %) and
aliphatic compounds (20 %–33 %), different from C1 (Fig. S10g). Overall, although
hydrolyzation-product pairs contributed to C1-assigned CHON+ formulas, the result
showed that oxidation is one of the key formation pathways for the abundant CHON
formulas assigned to HULIS components, especially C2. N-containing compounds, in
particular CHON compounds, substantially contributed a lot of light absorption of BrC
(Lin et al., 2017; Bluvshtein et al., 2017). The significant differences in the formation
processes of CHON compounds assigned to PARAFAC components may be an
important factor affecting the light absorption of BrC, even radiation forcing.



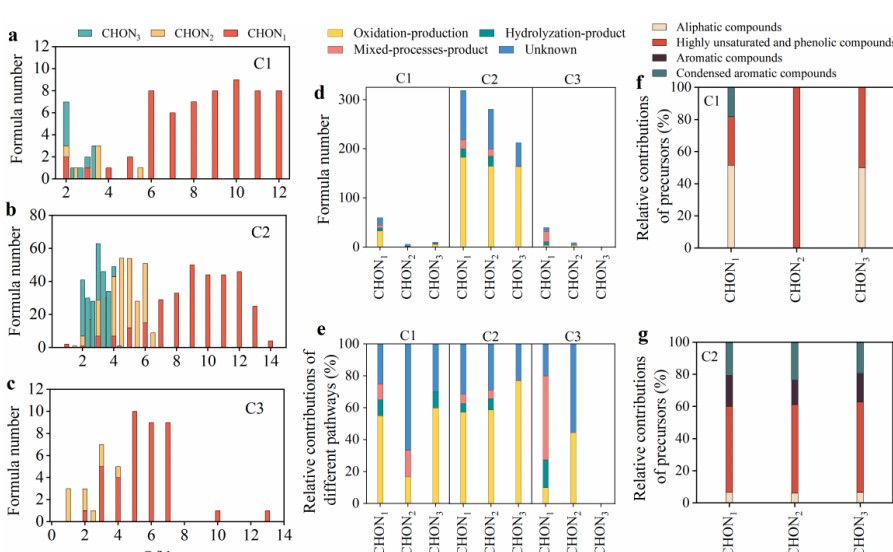


**Figure 4**. O/N ratios distribution of CHON– compounds assigned to C1 (a), C2 (b), and
C3 (c), and the corresponding formula number distribution of different pathways for CHON– molecules
assigned to C1–3 (d) and the relative contributions of different pathways (e). Indeed, panels (f–g)
refer to the relative contributions of different precursor types to the abundant oxidation-product pair
for $CHON_{1-3}$ of C1 and C2, respectively.

Multiple studies have demonstrated that S-containing compounds have little or no

contribution to BrC chromophores (Song et al., 2019; Zeng et al., 2021), yet knowledge
of the formation pathway of S-containing compounds assigned to PARAFAC
components could help us understand its formation mechanism with possible similar
pathways. Previous work investigated the prevalence of the epoxide formation pathway
for organosulfates (OS) and nitrooxy-OS by examining the presence of precursor-
product pairs of the CHOS (or CHONS) and the corresponding CHO (or CHON)
compounds (Lin et al., 2012b). If the epoxides form, both sulfate and water can act as
nucleophiles, and thus, both the OS and the corresponding alcohol should be present.
In our samples, although few S-containing compounds were assigned to C1 (19, 20 %
of its formulas) and C2 (56, 4.2 %) in that were less than C3 (117, 47 %), 84 %–100 %
of CHOS compounds assigned to PARAFAC components had O/S ratios > 4, and 15 %–
70 % of CHONS compounds had O/S > 7 for $CHON_1S$ compounds, and > 10 for
$CHON_2S$ compounds, which satisfied the epoxides pathway. In Table 1, a similar





analysis shows this process over CHOS and CHONS assigned to three PARAFAC
components. On average, for over 26 %–58 % of CHOS assigned to C1–3, the
corresponding CHO alcohol formulas were found, consistent with the epoxide
intermediate pathway for OS formation, yet the remaining 42 %–74 % of CHOS, have
no corresponding alcohols, may be formed from other pathways. In contrast, only a
small fraction of CHONS OS assigned to C1 and C2 were observed as exist for the
corresponding CHON alcohol formulas, yet this fraction increased to 47 % of CHONS
OS assigned to C3, implying that the epoxide pathway is more significant for the
formation of OS in the CHONS assigned to C3. In addition, previous work showed that
organonitrates hydrolyze more rapidly than OS (Hu et al., 2011). This hydrolysis
process could substitute the nitrooxy group with a hydroxyl group (i.e., $-HNO_3 + H_2O$).
On average for 76 % of CHOS assigned to C3 (Table 1), a corresponding CHONS is
present, which could be explained by the hydrolysis of nitrooxy groups of OS, yet not
common for CHOS assigned to C1 and C2. The prevalent hydrolysis of nitrooxy groups
of OS is commonly observed in the Pearl River Delta of China and Bakersfield,
California (Lin et al., 2012b; O'Brien et al., 2014). This collectively indicated that the
CHOS assigned to C3 mainly derived from the epoxide intermediate pathway and
hydrolysis process (80 %), and CHONS were mainly derived from the epoxide
intermediate pathway.
**Table 1**. Numbers and percentages of formation and hydrolysis reactions of CHOS and CHONS.

Number and Percentages of Precursor-Product Pairs

| Sample type | CHOS– $SO_3 \rightarrow$ CHO | CHONS– $SO_3 \rightarrow$ CHON | CHONS+OH– $NO_3 \rightarrow$ CHOS | References |
|---|---|---|---|---|
| Karachi, Pakistan | | | | |
| C1 | 11 (58 %) | 2 (22 %) | 0 (0.0 %) | |
| C2 | 12 (26 %) | 0 (0.0 %) | 2 (4.3 %) | This study |
| C3 | 26 (31 %) | 16 (47 %) | 63 (76 %) | |
| Bakersfield, California | | | | |
| Midnight to 6 A.M. | 77 (45 %) | 17 (30 %) | 43 (74 %) | |
| 6 A.M. to Noon | 70 (41 %) | 17 (27 %) | 53 (79 %) | (O'Brien et al., 2014) |
| Noon to 6 P.M. | 115 (63 %) | 12 (41 %) | 26 (75 %) | |
| 6 P.M. to Midnight | 131 (79 %) | 10 (30 %) | 22 (66 %) | |



| Guangzhou, China | | | | |
|---|---|---|---|---|
| Urban | 148 (65 %) | 2 (2.8 %) | 52 (75 %) | |
| Suburban | 74 (69 %) | 7 (15 %) | 22 (47 %) | (Lin et al., 2012b) |
| Rural | 113 (75 %) | 3 (5.1 %) | 38 (69 %) | |
| Urban | 699 (27 %) | 508 (19 %) | - | (Jiang et al., 2022a) |

Numbers and percent of CHOS and CHONS compounds with corresponding CHO, CHON, and
CHOS formulas in the same mass spectra for the following reactions: $C_xH_yO_zS \rightarrow C_xH_yO_{z-3} + SO_3$;
$C_xH_yO_zN_wS \rightarrow C_xH_yO_{z-3}N_w + SO_3$; $C_xH_yO_zN_wS \rightarrow C_xH_{y+1}O_{z-2}N_{w-1}S - H_2O + HNO_3$.
The remaining O-containing compounds, including CHO–, CHO+, and CHONa+
groups, are mainly substituted with multiple polar functionalities including carboxyl,
carbonyl, and hydroxyl groups. Except for the primary emission source for CHO
compounds, secondary formation was suggested to occur (Lee et al., 2014; Lin et al.,
2014). Kundu et al. (2012) detected hundreds of CHO molecules in SOA from limonene
ozonolysis. Specially, 42 % of CHO compounds (neutral molecules) assigned to C1,
34 % assigned to C2, and 5.3 % assigned to C3 were commonly detected in mass spectra
of limonene ozonolysis samples (Table S5), indicating a significant fraction of the
oxidation process for the formation of CHO compounds assigned to C1 and C2.
Undoubtedly, aging (e.g., $NH_3$) of limonene ozonolysis SOA is susceptible to the
formation of oligomeric products with extensive conjugation of π-bonds creating the
BrC chromophores (Laskin et al., 2014). In contrast, some CHO assigned to C2 was
observed in the naphthalene photooxidation products, biomass burning, and aqueous-
phase reactions of phenols (Table S6), yet not common for CHO assigned to C1 and C3.
Remarkably, a majority of CHO (about 90 %) assigned to C3 may be formed from the
other pathways. The $n_c$-OSc space of the CHO compounds assigned to C3 allowed for
a probable hydrocarbon-like OA and biomass-burning OA source for this group (Fig.
S11) (Kroll et al., 2011). Overall, the obtained formation pathways for PARAFAC
components based on their molecular composition may explain the molecular similarity
in C1 and C2 as described above. For example, 36 % of overlapped CHON compounds
in C1 and C2 formulas were observed as oxidation-product pair, and 73 % of overlapped
CHO compounds were detected in limonene ozonolysis SOA, suggesting that the





oxidation reaction may be an important reason for their molecular-level correlation of
C1 and C2.

### 3.4. Underlying implication of PARAFAC component to BrC absorption

The exact molecular identities of the BrC chromophores are expected to have a
high degree of conjugation across the molecular skeleton and large absorption cross-
sections (Lin et al., 2018). Fluorescence is a subset of BrC chromophores that absorb
certain wavelengths of light and re-emit a fraction of that energy. Thus, the components
decomposed by PARAFAC analysis represent a category of compounds that have
similar chemical properties, which can define as the chemical identification of BrC
chromophores. Our recent work found that the PARAFAC component with the longest-
emission maxima had the largest coefficient by fitting the light absorption and
PARAFAC components using multiple linear regression (Tang et al., 2021), indicating
that this component could significantly contribute to light absorption. However, the
molecular-level correlation between PARAFAC components and BrC chromophores
was not fully understood. Solving this gap could promote the application of the EEM-
PARAFAC method for studying BrC chromophores. Lin et al. (2018) proposed a plot
of carbon number versus DBE based on the formulas, and the compounds with DBE/C
ratios of 0.5~0.9 are potential BrC chromophores. The shaded area in Fig. 5 highlights
the molecules assigned to C1–3 matching this criterion. Of the formulas assigned to C1,
37 % of ESI– and 16 % of ESI+ formulas are located in the "BrC domain" marked by
the brown area. In contrast, a larger number of C2 formulas are located in this region,
accounting for 65 % and 31 % of total C2 formulas detected in the ESI– and ESI+,
respectively, yet only 12 % of ESI– and 21 % of ESI+ formulas assigned to C3 are
within "BrC domain". The result implied that a category of compounds that produce
C2 fluorophores could substantially contribute to the light absorption of BrC, as further
confirmed by the highly overlapped formulas between C2 and BrC-assigned formulas
(72 %–94 % of C2 formulas and 47 %–57 % of BrC formulas) (Fig. S9). This may be
the molecular basis for the tight correlation between C2 and $MAE_{365}$ observed in this
study. Undoubtedly, the $NO_x$ addition reaction generally produced highly absorbing
substances (Li et al., 2020a; He et al., 2022; Siemens et al., 2022), as this process largely
occurred in the formation of C2, suggesting that the molecular consistency between C2
and BrC may be due to the $NO_x$ addition reaction. Remarkably, some of the molecules
assigned to PARAFAC components were observed as not matching this region, perhaps
because they underwent similar processes like BrC chromophores. In general, the
statistical significance of correlation provides more intrinsic information for
understanding the molecular basis and fate of atmospheric PARAFAC components,
even to BrC chromophores.

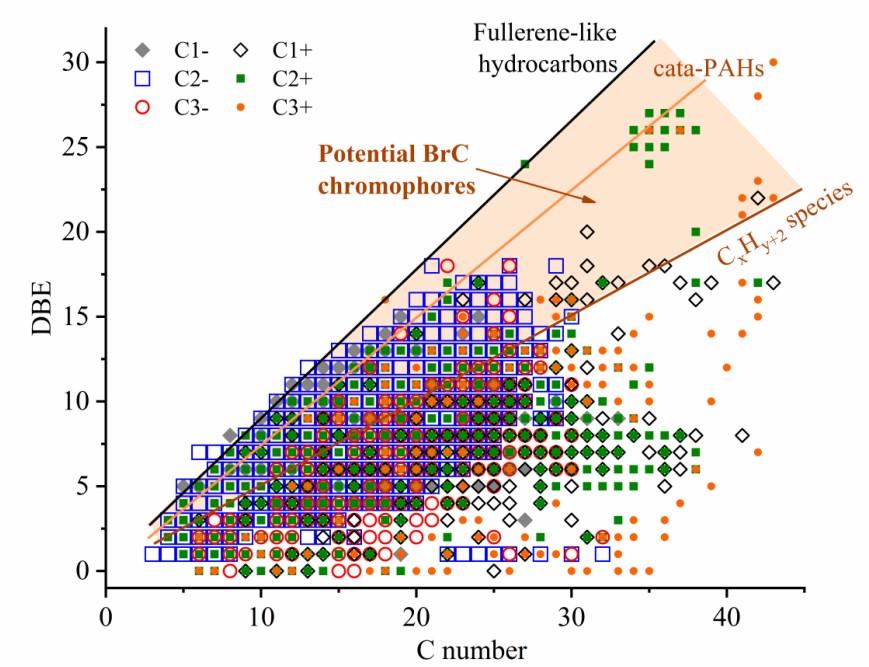


**Figure 5**. Plot of the double bond equivalent (DBE) vs the number of carbon atoms in PARAFAC
components-assigned molecular formulas. Lines indicate DBE reference values of linear conjugated
polyenes $C_xH_{x+2}$ with DBE = 0.5 × c (brown solid line), *cata*-condensed PAHs (yellow solid line),
and fullerene-like hydrocarbons with DBE = 0.9 × c (black solid line). Data points inside the orange-
shaded area are potential BrC chromophores (Lin et al., 2018).

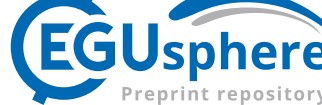

**4. Conclusions and atmospheric implications**


In this study, the fluorescence and molecular compositions of water extracts in
Karachi aerosol were characterized using EEM spectroscopy and negative and positive
ESI FT-ICR MS, respectively. The identified two humic-like (C1 and C2) and one
protein-like (C3) components are commonly observed in the atmosphere, which has
different molecular characteristics and formation pathways. The characteristics of
fluorescent components may be indicative of the sources, atmospheric processes, and
reactivity of water-soluble aerosol components. Of what we observed in this study, C1
and C2 were enriched in N-containing compounds, yet C2 is more associated with
higher aromaticity, higher N content, and highly oxygenated compounds; C3 is
characterized as low conjugation, oxygen-depleted, N-depleted, but S-enriched species.
Previous studies showed that fluorescent components may be employed as source
indicators for OA (Chen et al., 2016b; Tang et al., 2020). Aromatic compounds
generally originate from combustion emissions; fluorescent components with high
aromatic moieties, such as the C2 component, may be derived from anthropogenic
precursors that have experienced high oxidation (high NOSC and $\overline{OS_C}$). However, the
feature of C3 may originate from primary emissions, e.g., vehicular exhaust (Mladenov
et al., 2011).
Oxidation formation pathway was observed as an important process for the
formation of C1 and C2, especially their assigned CHON compounds. The O/C ratios
of C1- and C2-assigned ESI– compounds (0.48±0.18 and 0.59±0.20, respectively) also
exhibit higher values, especially C2, than that from primary emissions and some SOA,
as shown in Fig. 6. This molecular-level character of C2 indicates that this component
may be used as a secondary source tracer, consistent with a recent study observed using
online EEM monitoring (Chen et al., 2021b). The secondary information involved could
be used to probe the secondary source of BrC, which is poorly understood because of
its chemical complexity. In addition, C2 with the longest emission maxima (~494 nm),
was assigned to a large number of compounds that matched the "potential BrC" region



and overlapped with BrC-associated formulas, and readily correlated tightly with
$MAE_{365}$, collectively indicating that a category of compounds illuminating C2 may
significantly contribute to the BrC light absorption, which also observed in our previous
study (Tang et al., 2021). In the study of Chen et al. (2019), they showed that almost all
DTT activity is attributed to the C7 chromophore (99%), a component similar to C2 in
this study. Given the high light absorption radiation and health effect induced by C2,
much attention should be drawn to further study. It is also suggested that the commonly
used fluorescence characteristics derived from aquatic environments may not be
applicable, as references, to atmospheric WSOC study (refer to Text S7). Our findings
on the molecular compositions and formation mechanisms of atmospheric fluorescent
components are expected to be helpful to further studies using the EEM-PARAFAC as
a tool to study atmospheric BrC (Laskin et al., 2015).



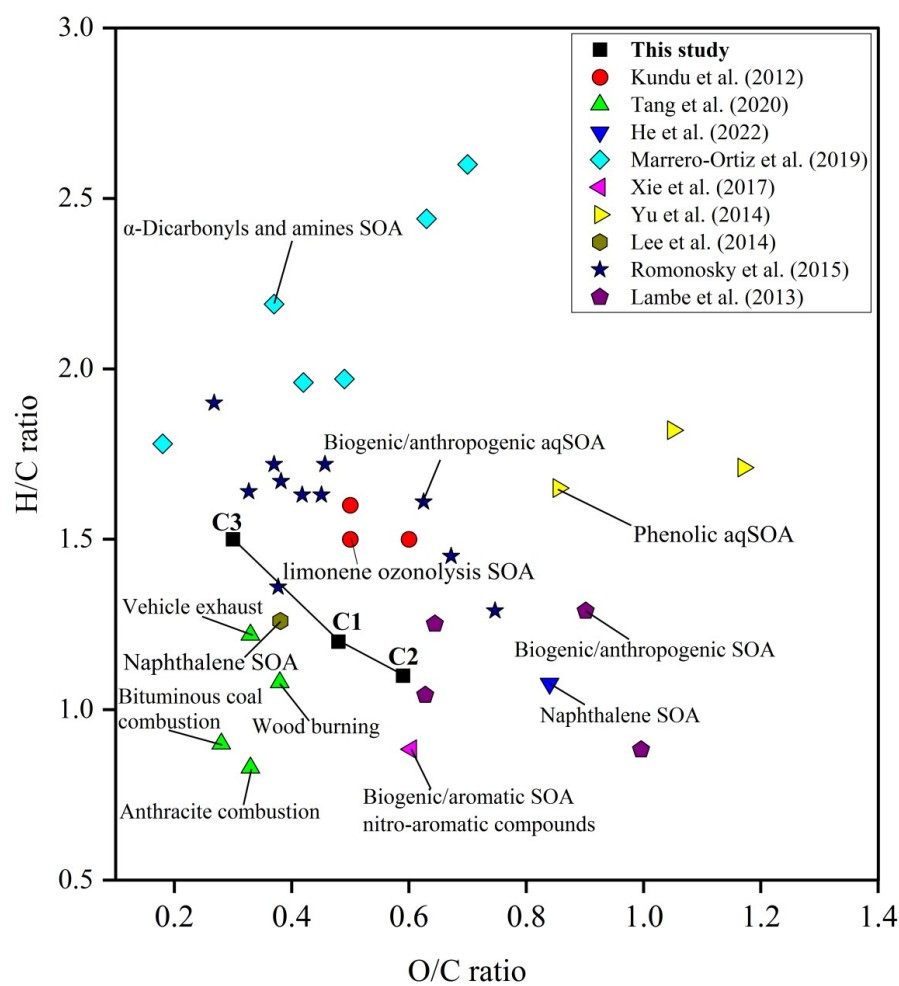

**Figure 6**. Comparison of O/C and H/C ratios of C1–3 assigned ESI– compounds in water-soluble
organic compounds in this study with other primary emissions and secondary organic aerosol.



**Data availability.** The data used in this study are available in the Harvard
Dataverse (https://doi.org/10.7910/DVN/RWIJZT, Tang, 2023).
**Supplement.** The supplement related to this article is available online.
**Author Contributions**. GaZ and JT designed the study. JH provided the samples.
JT, YM, HJ, ZZ, and BJ carried out the analysis. JianT provided the instrument. JT
processed the data and wrote the original draft. JL, SZ, GuZ, YC, CT, JS, and GaZ
review the manuscript.
**Competing interests**. The authors declare that they have no conflict of interest.
**Acknowledgments.** We appreciate Boji and Yangzhi for their help with the model
and FT-ICR MS analysis.
**Financial support.** This research has been supported by the National Natural
Science Foundation of China (Grant nos. 42030715 and 42207308), the Alliance of
International Science Organizations (ANSO-CR-KP-2021-05), the Guangdong Basic
and Applied Basic Research Foundation (2017BT01Z134, 2021A0505020017, and
2023B1515020067), and Youth Innovation Promotion Association, CAS (2022359).

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
