# Peer review of "(PM) water-soluble chromophores from Karachi (Pakistan) over 4 South Asia"

_EGUsphere, 2023_

## Referee Comment (RC1)

**Summary**

In this preprint, the authors utilized excitation-emission matrix (EEM) fluorescence spectroscopy and Fourier transform ion cyclotron resonance mass spectrometry (FT-ICR MS) to examine the molecular composition and formation mechanisms of water-soluble organic carbon (WSOC) in atmospheric aerosols gathered from Karachi, Pakistan. They identified three parallel factor (PARAFAC) components, consisting of two humic-like (C1 and C2) and one protein-like (C3). They further investigated the connection between PARAFAC components and possible brown carbon (BrC) fluorescent species, along with absorbance characteristics, which could aid in broadening our knowledge of BrC in the atmosphere. With a few adjustments, clarifications, and a thorough language revision for clarity, this study has the potential to contribute to the field and be suitable for publication in ACP.

**Strengths**

The authors leveraged a blend of advanced analytical techniques, such as EEM spectroscopy and FT-ICR MS, to characterize WSOC in aerosols.

By identifying three PARAFAC components and examining their molecular relationships, the study provides a more in-depth understanding of the formation processes and properties of fluorescent species in the atmosphere.

The insights gained on the molecular compositions and formation mechanisms of atmospheric fluorescent components can prove valuable for future investigations that employ EEM-PARAFAC to study atmospheric BrC.

**Areas for improvement**

The study focuses on aerosols from Karachi, Pakistan, so the conclusions might not represent other geographical areas. Although there are comparisons with results from Bakersfield and Guangzhou (among others), it would be beneficial for the authors to critically discuss the broad applicability of their overall findings to other locations in the conclusion, or suggest future studies using their methodologies to include aerosols from diverse regions.

In Section 3.3, the authors attempt to explain the differences in oxidation pathways (depending on precursor types) for the formation of C1 and C2 components. It would be helpful if the authors could elucidate this explanation (Lines 520-522) and incorporate a more detailed summary in both the conclusion (Lines 586-587) and abstract.

Although the study identifies molecular families associated with each of the three components, their specific molecular formulas are not detailed, potentially limiting the study's replicability. Providing more information on abundant (in terms of MS signal intensity) molecular formulas associated with each component would allow other researchers to expand upon these findings. In addition, based on Lines 537-540 (Section 3.4) and Figure 5, the reviewer is particularly interested in knowing the molecular formulas of major (potential) BrC species (nitro-aromatics, CHO species) within the C2 component.

Line 562: It may be worthwhile to include an alternate version of this figure that represents the number of carbon and nitrogen atoms (C+N) on the *x*-axis, and discuss any discrepancies with the current version. This alternate figure might be more pertinent to BrC, as nitroaromatic groups (if present) in CHON are potential chromophores. When considering C+N, the data points for N-containing compounds in this figure would shift to the right.

Several sentences are difficult to understand, and their meanings are unclear. The authors should rectify any remaining language issues to enhance the manuscript's readability. After these improvements, another review might be required to ensure the precision of all statements in the revised manuscript.

**Additional comments and suggested edits**

Lines 1-3: Consider rearranging the title for clarity: "Molecular signatures and formation mechanisms of water-soluble chromophores in particulate matter from Karachi (Pakistan) in South Asia."

Line 56: Change to "originates."

Line 58: Change to "formation processes, such as aqueous-phase reactions from anthropogenic or biogenic emissions."

Lines 60-61: Change to "Light-absorbing organic (or brown) carbon (BrC) is an important component of WSOC…"

Line 74: Change to "...absorb light at certain wavelengths..."

Lines 115-116: Could you elaborate on whether primary or aged biomass burning emissions are anticipated and how they might influence the interpretation of BrC results?

Line 126: Change to "Total suspended particulate matter (TSP) was collected..."

Line 148: Change to "PARAFAC."

Line 285: Please clarify what the error bars signify in panels (c) and (d).

Lines 310-312: Could the authors provide insights as to whether the molecular formulas that showed no correlation with any PARAFAC component are predominantly aliphatic and sugar compounds, which are not typically expected to exhibit light absorption or fluorescence?

Line 362: Change to "saturated."

Line 364: Change to "lower."

Line 604: Summarize Text S7 briefly.

Comments on the SI

Text S3: Please explain why liquid chromatography was not employed for the separation of compounds prior to the MS analysis, and discuss any potential effects this decision might have on the current study.

Lines 149 & 154: Please elucidate the distinction between the carbon oxidation state and the nominal oxidation state of carbon. Also, provide a rationale for the application of both these metrics in this manuscript.

Line 209: Remove "using."

Line 232: Remove the first "that."

Line 233: Change to "may not have the same character as in..." or similar.

Lines 249-251: Rewrite the sentence for clarity.

Line 380: Correct "streams" in the figure legend.

---

## Referee Comment (RC2)

**Summary:**

In this scientific work, the authors have used excitation-emission matrix (EEM) fluorescence spectroscopy coupled with Fourier transform ion cyclotron resonance mass spectroscopy (FT-ICRMS) to study the chemical composition of water soluble organic carbon (WSOC) ambient aerosols in Karachi, Pakistan. In addition to that, they tried to postulate the different formation mechanisms of these organic compounds. They identified three broad components of moieties using parallel factor analysis (PARAFAC), of which two were humic-like (C1 and C2) and one was protein like (C3). They also tried to correlate these PARAFAC components to the extensive datasets of brown carbon (BrC) chromophores available in literature through the total number of carbon (C) atoms present in the molecular formula and double-bond equivalence (DBE). Except the discussion of formation mechanism, the rest of the methodology and findings of this work is quite trivial and another repetition of multiple EEM spectroscopy based ambient aerosol characterization studies.

**Strength of this work:**

The authors analysed the FT-ICR MS data and came up with the most probable formation mechanisms from different molecular signals in addition to EEM spectroscopic studies.

**Limitations of this work:**

- *Lack of discussion on seasonal variation:* As seen in Figure 1, the study site is located at a very interesting geographical location with great seasonal variation in wind direction. In pre-monsoon and monsoon, the wind flow is directed from middle-east Asia and Arabian Sea, whereas the wind trajectories arise from North Pakistan and North-West India during post-monsoon and winter. This will result in very different chemical compositions of ambient particulate matter (PM) reaching the study site. For example, in the months of May-June, the aerosol composition will be close to marine aerosol composition, but in Oct-Dec there will be molecular signals of biomass burning emissions as previous studies have pointed out extensive crop burning and biomass burning for heat generation during winter in that part of the world. Which suggests that the wintertime aerosols will probably have higher S content and less oxygenated organics because the OH radical photochemistry is limited during winter and the atmospheric transformation is driven by NOx chemistry. The authors have reported all the data in a combined way, which does not give the readers the broader picture of the regional specific atmospheric chemistry of the study site. Postulating molecular formation pathways without considering the meteorological conditions can also lead to erroneous assumptions.

- *Lack of relevant references:* In continuation to the previous point, as the authors have not discussed the geographical context of this work, they have also failed to compare their findings with previous works carried out in similar locations. Although they have mentioned a few studies carried out in the Indo-Gangetic Plane (IGP) while reporting mass absorption efficiency (MAE) and Aerosol absorption exponent (AAE), this kind of comparisons have not been made for EEM spectroscopic studies. Previous EEM spectroscopic analyses of ambient aerosol in IGP and other parts of the world have found similar PARAFAC components (two HULIS and one protein like). These references from around the globe should be mentioned and compared with the findings of this work.

- *Structure of the Results and Discussion Section*: The results and discussions section needs to be restructured. For the convenience of the readers, the discussion of "Underlying implication of PARAFAC component to BrC absorption" should be done under section 3.3 and the discussion on formation pathways should be under section 3.4. That way the flow of information will be more coherent.

- The discussion on formation pathway of S-containing compounds should be more condense and can also be moved to supplementary information (SI). The authors have mentioned that S containing compounds have almost no effect on BrC chromophores. They also reported that the two Humic-like PARAFAC components had very little S containing compounds, mostly component C3 had the highest S containing compound. Eventually it has also been shown that component C3 has the lowest overlap with the BrC region in figure 5. Therefore, in terms of climate relevant BrC chromophores, this pathway is not as important as CHO and CHON formation pathways.

- The overall grammar and clarity of the current section 3.3 (proposed to be made section 3.4 after restructuring) is unsatisfactory. Too much statistics have been used in sentences, which can instead be represented graphically. Sentences need to be written with proper grammar, for example 407-408 is unclear and needs to be rewritten, so does 413. There are many occurrences like this, so the reviewer suggests a rewriting of this whole section in a clear and concise manner. In the rewriting, the authors should also elaborate and clarify how they reached conclusions made in 520-522.

  - **But most importantly in this section**, the authors have looked at some molecular signals found through FT-ICR MS and compared those molecules with aged byproducts of certain precursors reported in previous literature. And by doing that they have tried to postulate these known reaction pathways to be present in their study samples. These are good hypotheses, but there is no concrete way of establishing these reaction pathways in the scope of this study. Therefore, it doesn't add up as novel new information. If the authors can combine other analytical study with the collected filters (if there is any remaining), for example $^{1}$H NMR spectroscopy of WSOC, in which they can quantitatively compare the spectra of the precursor and aged molecules with their sample spectra and confirm its presence, that would be a much stronger argument for the formation pathways.

**Minor corrections:**

Title: The title should be reconsidered. Instead of water soluble particulate matter, water soluble organic carbon is a preferable choice. The study location should be kept Karachi or mentioned a location in South East Asia.

**Corrections in SI:**

**Figure S2:** The title of the figure mentions February 16, but the legend in the figure shows the trajectory starting from 17 February 2016.

---

## Author Comment (AC1)

Thanks to the anonymous reviewer for their constructive comments on the manuscript and helpful suggestions for further improvement. Please find detailed responses below in blue-color font.

**Response to Anonymous Referee #1**

**Summary**
In this preprint, the authors utilized excitation-emission matrix (EEM) fluorescence spectroscopy and Fourier transform ion cyclotron resonance mass spectrometry (FT-ICR MS) to examine the molecular composition and formation mechanisms of water-soluble organic carbon (WSOC) in atmospheric aerosols gathered from Karachi, Pakistan. They identified three parallel factor (PARAFAC) components, consisting of two humic-like (C1 and C2) and one protein-like (C3). They further investigated the connection between PARAFAC components and possible brown carbon (BrC) fluorescent species, along with absorbance characteristics, which could aid in broadening our knowledge of BrC in the atmosphere. With a few adjustments, clarifications, and a thorough language revision for clarity, this study has the potential to contribute to the field and be suitable for publication in ACP.

Response: Thank you for appreciating our work and providing valuable suggestions. We have obtained professional language editing and have revised the manuscript according to your comments. The revised sections are highlighted for easy reference.

**Strengths**
The authors leveraged a blend of advanced analytical techniques, such as EEM spectroscopy and FT-ICR MS, to characterize WSOC in aerosols.

By identifying three PARAFAC components and examining their molecular relationships, the study provides a more in-depth understanding of the formation processes and properties of fluorescent species in the atmosphere.

The insights gained on the molecular compositions and formation mechanisms of atmospheric fluorescent components can prove valuable for future investigations that employ EEM-PARAFAC to study atmospheric BrC.
Response: We are grateful to the reviewer for recognizing our work and providing valuable suggestions. The suggestions would help us improve the manuscript.

**Areas for improvement**

The study focuses on aerosols from Karachi, Pakistan, so the conclusions might not
represent other geographical areas. Although there are comparisons with results from
Bakersfield and Guangzhou (among others), it would be beneficial for the authors to
critically discuss the broad applicability of their overall findings to other locations in
the conclusion, or suggest future studies using their methodologies to include aerosols
from diverse regions.

Response: Thanks. This study investigated the molecular compositions and formation
pathways of water-soluble PARAFAC components in this interesting geographical
location in Karachi, Pakistan. Our findings can potentially be applied to other locations,
highlighting two main aspects. First, we obtain the molecular characteristics of
PARAFAC components that are consistent with their emission wavelength maximum.
The trend of DBE, O/C, $\overline{OS}_C$, and $AI_{mod}$ of molecular compounds assigned to C1–3 is
as follows: C2 (494 nm) > C1 (415 nm) > C3 (337 nm). Similar patterns were observed
in the molecular formulas of humic-like components compared to those of protein-like
components in Guangzhou, a city in southern China by He et al. (2023). These findings
suggest that the maximum emission wavelength of fluorescent components may serve
as an indicator of the chemical characteristics of BrC chromophores, such as the level
of unsaturation and oxidation. Second, the pathway of oxidation formation was
identified as an important process for the formation of C1 and C2. This is specific to
their assigned CHON compounds. The deduced $N_2O_5$ oxidation reaction, which formed
organic nitrates, accounts for a significant fraction of CHON compounds assigned to
C1 and C2. While C2 has more precursors of aromatic/condensed aromatic compounds
for CHON compared to C1, suggesting that this component contains more nitro-
aromatic compounds. Furthermore, C2 is assigned a large number of compounds that
match the "potential BrC" region and overlap with BrC-associated formulas and
exhibits a close correlation with $MAE_{365}$. Taken together, these findings imply that a
group of compounds that contribute to C2 fluorescence may significantly contribute to
the light absorption of BrC. According to these findings, we inferred that this
component likely represents a class of strongly absorbing substances, specifically nitro-
aromatic compounds. More discussions were shown in lines 820-897 in the revised
manuscript.

In Section 3.3, the authors attempt to explain the differences in oxidation pathways
(depending on precursor types) for the formation of C1 and C2 components. It would
be helpful if the authors could elucidate this explanation (Lines 520-522) and
incorporate a more detailed summary in both the conclusion (Lines 586-587) and
abstract.

Response: We draw the conclusion (original version in lines 520-522) based on the
elemental compositions of the overlapping molecules assigned to C1 and C2, which
consist mainly of CHON and CHO compounds. Additionally, the formation pathways
of these overlapping molecules were found to be a significant part of oxidation-derived
pathways, with CHON compounds primarily attributed to $N_2O_5$ oxidation and the CHO
compounds were highly detected in the SOA formed from limonene ozonolysis. For the
revised sections, please refer to lines 763-778 in the revised manuscript.
In addition, the detailed summary was further discussed in the conclusion (lines 849-
857) and mentioned in the abstract (lines 46-50).

Although the study identifies molecular families associated with each of the three
components, their specific molecular formulas are not detailed, potentially limiting the
study's replicability. Providing more information on abundant (in terms of MS signal
intensity) molecular formulas associated with each component would allow other
researchers to expand upon these findings. In addition, based on Lines 537-540 (Section
3.4) and Figure 5, the reviewer is particularly interested in knowing the molecular
formulas of major (potential) BrC species (nitro-aromatics, CHO species) within the
C2 component.
Response: In fact, the data on correlation analysis is accessible on Harvard Dataverse
(https://doi.org/10.7910/DVN/RWIJZT, Tang, 2023) in the data availability section for
the convenience of other researchers. The reader can easily obtain the detailed
molecular formulas that were assigned to each PARAFAC component (or $Abs_{365}$,
optical indices, and fluorescence indices) in lines 903-904 in the revised manuscript.
In addition, we have revised the plot and incorporated subgroups of potential BrC
compounds, illustrated in the new Figure 4. It was found that CHO and CHON
contributed a significant portion (98 %) of C2-assigned formulas located in the "BrC
domain", with CHON accounting for 68% of them. Furthermore, 80% of potential BrC
CHON compounds assigned to C2 have O/N ≥ 3, indicating that they are species of
nitro-aromatics. For more detailed information, please refer to the Harvard Dataverse
(https://doi.org/10.7910/DVN/RWIJZT, Tang, 2023). The updated discussions were
presented in lines 544-557 in the revised manuscript.

[Figure]

**Figure 4**. Plot of the double bond equivalent (DBE) vs the number of C + N atoms of PARAFAC components-assigned molecular formulas, and the corresponding subgroups of potential BrC components in the ESI– (a, c) and ESI + (b, d) modes. Lines indicate DBE reference values of linear conjugated polyenes $C_xH_{x+2}$ with DBE = $0.5 \times C$ (brown solid line), *cata*-condensed PAHs (yellow solid line), and fullerene-like hydrocarbons with DBE = $0.9 \times C$ (black solid line). Data points inside the lines are potential BrC chromophores (Lin et al., 2018).

Line 562: It may be worthwhile to include an alternate version of this figure that represents the number of carbon and nitrogen atoms (C+N) on the x-axis, and discuss any discrepancies with the current version. This alternate figure might be more pertinent to BrC, as nitroaromatic groups (if present) in CHON are potential chromophores. When considering C+N, the data points for N-containing compounds in this figure would shift to the right.

Response: Thank you for your suggestions. We have revised this plot by adding a subgroup plot of the potential BrC components assigned to C1−3 (Figure 4). The DBE values of CHON and CHONS compounds shift right by 1-4 units due to the sum number of carbon and nitrogen atoms. The new discussions were presented in lines 544-557 in the revised manuscript.

Several sentences are difficult to understand, and their meanings are unclear. The
authors should rectify any remaining language issues to enhance the manuscript's
readability. After these improvements, another review might be required to ensure the
precision of all statements in the revised manuscript.
Response: We apologize for the low quality of the language in our manuscript. We spent
a long time revising the manuscript, which involved repeatedly adding and removing
sentences and paragraphs. This led to a significant decrease in readability. We have
worked on improving both the language and the readability of the document.
Additionally, we have invited fellow experts to make professional language editing for
the manuscript. We hope that the clarity and language sophistication has been
significantly improved.
**Additional comments and suggested edits**
Lines 1-3: Consider rearranging the title for clarity: "Molecular signatures and
formation mechanisms of water-soluble chromophores in particulate matter from
Karachi (Pakistan) in South Asia."
Response: Thank you. The title has been modified according to your recommendations.
Line 56: Change to "originates."
Response: We have revised it (in line 65).
Line 58: Change to "formation processes, such as aqueous-phase reactions from
anthropogenic or biogenic emissions."
Response: We have revised it (in lines 67-68).
Lines 60-61: Change to "Light-absorbing organic (or brown) carbon (BrC) is an
important component of WSOC…"
Response: We have revised it (in lines 69-71).
Line 74: Change to "...absorb light at certain wavelengths..."
Response: We have revised it (in line 85).
Lines 115-116: Could you elaborate on whether primary or aged biomass burning
emissions are anticipated and how they might influence the interpretation of BrC results?
Response: Of course. In the revised paragraph, nss-ndust-$K^+$ (the fraction unrelated to
sea salt and mineral dust, nss-ndust) was used as a marker of biomass burning to
investigate its influences (Pio et al., 2007; Zhou et al., 2019). A good linear correlation was found between $Abs_{365}$ (a proxy for BrC) and nss-ndust-$K^+$ (r = 0.57, p < 0.01, Fig.

S5) during non-monsoon seasons. In contrast, the absence of a linear relationship between $Abs_{365}$ and nss-ndust-$K^+$ in the monsoon season (Fig. S5) indicates no influence of biomass burning emissions. Consistently, $MAE_{365}$ was higher during non- monsoon seasons than the monsoon season, which suggests that biomass burning has an important influence on water-soluble BrC during non-monsoon seasons. Please refer to lines 366-372 in the revised manuscript.

[Figure]

**Figure S5.** Correlation between nss-ndust-$K^+$ and $Abs_{365}$ in non-monsoon and monsoon seasons.

Line 126: Change to "Total suspended particulate matter (TSP) was collected..."

Response: We have revised it (in line 155).

Line 148: Change to "PARAFAC."

Response: We have revised it (in line 184).

Line 285: Please clarify what the error bars signify in panels (c) and (d).

Response: The error bars shown in panels c and d represent the standard deviation. We have added after the plot in lines 388-389 in the revised manuscript.

Lines 310-312: Could the authors provide insights as to whether the molecular formulas that showed no correlation with any PARAFAC component are predominantly aliphatic and sugar compounds, which are not typically expected to exhibit light absorption or
fluorescence?

Response: Of course. The molecular formulas that did not correlate with any of the
PARAFAC components are predominantly aliphatic compounds and highly unsaturated
and phenolic compounds (a total of 87% of the unassigned molecules in the ESI– mode
and 83% in the ESI+ mode, respectively). Please refer to lines 417-420 in the revised
manuscript.

Line 362: Change to "saturated."
Response: We have revised it (in line 485).

Line 364: Change to "lower."
Response: We have revised it (in line 488).

Line 604: Summarize Text S7 briefly.
Response: Thank you for your suggestions. We have revised it as follows: "Furthermore,
the degree of molecular similarity between fluorescent components in various systems
was examined. Despite the similarity of fluorescent components in the atmosphere and
aquatic environment, their respective molecular formulas have substantial differences.
The fluorescence characteristics derived from aquatic environments may not be suitable
as references for studying atmospheric WSOC, as discussed in Text S8." Please refer to
lines 887-893 in the revised manuscript.

**Comments on the SI**
Text S3: Please explain why liquid chromatography was not employed for the
separation of compounds prior to the MS analysis, and discuss any potential effects this
decision might have on the current study.

Response: This is because our aerosol samples were extracted using ultrapure water,
while the bulk samples were measured directly using a fluorometer (Aqualog; Horiba
Scientific, USA). Therefore, maintaining consistency of the corresponding fraction is
crucial to obtain the molecular formulas using FT-ICR MS.

Your suggestions are very valuable. If we choose liquid chromatography to separate
compounds before MS analysis, we must simultaneously obtain the EEM spectra of the
separated fractions, such as the method described by Lin et al. (2015; 2016). They
combine HPLC-PDA-HRMS to simultaneously obtain the light absorption and
molecular compositions of BrC chromophores. In addition, this proposal provides new
knowledge for investigating the molecular compositions of fluorescent components. It is unavoidable that analyzing bulk samples will result in some missed underlying
information. For example, Spranger et al. (2019) used a 2D-liquid chromatographic
fractionation method, coupled with direct infusion electrospray ionization FT-ICR MS,
and observed a 2.3-fold increase in the number of molecular formulas detected in the
fractionated sample (18144) compared to bulk sample analysis without fractionation
(7819). So that will be our focus in the future.
References:

Lin, P., Aiona, P. K., Li, Y., Shiraiwa, M., Laskin, J., Nizkorodov, S. A., and Laskin, A.: Molecular
Characterization of Brown Carbon in Biomass Burning Aerosol Particles, Environ. Sci. Technol.,
50, 11815-11824, https://doi.org/10.1021/acs.est.6b03024, 2016.
Lin, P., Laskin, J., Nizkorodov, S. A., and Laskin, A.: Revealing Brown Carbon Chromophores
Produced in Reactions of Methylglyoxal with Ammonium Sulfate, Environ. Sci. Technol., 49,
14257-14266, https://doi.org/10.1021/acs.est.5b03608, 2015.
Spranger, T., Pinxteren, D. V., Reemtsma, T., Lechtenfeld, O. J., and Herrmann, H.: 2D Liquid
Chromatographic Fractionation with Ultra-high Resolution MS Analysis Resolves a Vast Molecular
Diversity of Tropospheric Particle Organics, Environ. Sci. Technol., 53, 11353-11363,
https://doi.org/10.1021/acs.est.9b03839, 2019.

Lines 149 & 154: Please elucidate the distinction between the carbon oxidation state
and the nominal oxidation state of carbon. Also, provide a rationale for the application
of both these metrics in this manuscript.
Response: These two parameters represent the oxidation state of compounds. The
nominal oxidation state of carbon (NOSC) is estimated from the chemical formula by
assuming that all other elements are in their initial oxidation states (H = +1, O = −2, N
= −3, and S = −2) using the following equation (1) (Riedel et al., 2012), which is
commonly used in aquatic and soil systems.

$$\text{NOSC} = 4 - [(4c + h - 3n - 2o - 2s) / c] \quad (1)$$

The carbon oxidation state ($\overline{OS}_C$) is calculated using equation (2) (Kroll et al., 2011),
which is commonly used for atmospheric aerosols.

$$\overline{OS}_C = -\Sigma_i \, os_i \frac{n_i}{n_c} \quad (2)$$

Where, $OS_i$ is the oxidation state associated with element $i$, and $n_i/n_C$ is the molar ratio
of element $i$ to carbon. Furthermore, if we choose the initial oxidation states of H, O,
N, and S atoms, equation (2) is equal to equation (1), as used in the study by Hettiyadura et al. (2021) ($\overline{OS}_C = \frac{2o}{c} + \frac{3n}{c} + \frac{2s}{s} - h/c$). However, predicting the average nitrogen oxidation state is a very difficult or even impossible task. It is known that highly
oxidized organic nitrates as well as low oxidized amines, amino acids, and other N-
containing organics such as imidazoles and other N-heterocyclic compounds are present
in aerosol samples. Spranger et al. (2019) assumed different oxidation states for N and
S atoms, resulting in different $\overline{OS}_C$ values. To avoid this error, we choose $\overline{OS}_C$ to
describe the oxidation state of the compounds, and make assumptions about the $OS_i$ of
the different atoms: $OS_O = -2$; $OS_H = 1$; $OS_N = 0$; $OS_S = 0$, which refers to the study by
Spranger et al. (2019). Thus, the carbon oxidation state ($\overline{OS}_C$) of individual compounds
was calculated as $\overline{OS}_C = 2 \cdot O/C - H/C$.

References:

Hettiyadura, A. P. S., Garcia, V., Li, C., West, C. P., Tomlin, J., He, Q., Rudich, Y., and Laskin, A.:
Chemical Composition and Molecular-Specific Optical Properties of Atmospheric Brown Carbon
Associated with Biomass Burning, Environ. Sci. Technol., 55, 2511-2521,
https://doi.org/10.1021/acs.est.0c05883, 2021.
Riedel, T., Biester, H., and Dittmar, T.: Molecular Fractionation of Dissolved Organic Matter with
Metal Salts, Environ. Sci. Technol., 46, 4419-4426, https://doi.org/10.1021/es203901u, 2012.
Spranger, T., Pinxteren, D. V., Reemtsma, T., Lechtenfeld, O. J., and Herrmann, H.: 2D Liquid
Chromatographic Fractionation with Ultra-high Resolution MS Analysis Resolves a Vast Molecular
Diversity of Tropospheric Particle Organics, Environ. Sci. Technol., 53, 11353-11363,
https://doi.org/10.1021/acs.est.9b03839, 2019.

Line 209: Remove "using."
Response: We have revised it (in line 249).

Line 232: Remove the first "that."
Response: We have revised it (in line 312).

Line 233: Change to "may not have the same character as in..." or similar.
Response: We have revised it (in line 315).

Lines 249-251: Rewrite the sentence for clarity.
Response: We have rewritten it as follows: …whereas C1 formulas (in lakes) grouped
in aromatic compounds region. This implies that although some PARAFAC
components have similar fluorescence patterns and excitation/emission maximum, their
chemical properties vary in different environments. For instance, the degree of unsaturation in environment can affect them. Please refer to lines 334-339 in the revised
supplement.
Line 380: Correct "streams" in the figure legend.
Response: We apologize for this error. We have corrected the plot as follows.

[Figure]

**Figure S15.** (**a**) Comparison of the average chemical characteristics of molecules assigned to C1,
C2, and C3, as well as the optical indices of WSOC in the aerosols (marked as red dot), DOM from
rivers and streams (brown triangle, fluorescent component P1−6) and lakes (green diamond,
fluorescent component C1−6, and FI, BIX, HIX, SUVA, $A_{254}$, and $S_{250-600}$); (**b**) the corresponding
excitation and emission maxima in the EEM spectra (Stubbins et al., 2014; Kellerman et al., 2015).
Note that the fluorescent components associated with molecular data (**a**) of DOM in rivers and
streams were re-extracted from their supplement and the error bars were not shown. The error bars
in this study are shown in Table S5. The range of the red circle, green circle, and blue circle (**b**)
represent the range of HULIS-2, HULIS-1, and PLOM fluorescence, respectively. The arrows
indicate polymerization and degradation defined according to the study (Chen et al., 2016).

---

## Author Comment (AC2)

Thanks to the anonymous reviewer for their constructive comments on the manuscript and helpful suggestions for further improvement. Please find detailed responses below in blue-color font.

**Response to Anonymous Referee #2**

**Summary:**

In this scientific work, the authors have used excitation-emission matrix (EEM) fluorescence spectroscopy coupled with Fourier transform ion cyclotron resonance mass spectroscopy (FT-ICRMS) to study the chemical composition of water soluble organic carbon (WSOC) ambient aerosols in Karachi, Pakistan. In addition to that, they tried to postulate the different formation mechanisms of these organic compounds. They identified three broad components of moieties using parallel factor analysis (PARAFAC), of which two were humic-like (C1 and C2) and one was protein like (C3). They also tried to correlate these PARAFAC components to the extensive datasets of brown carbon (BrC) chromophores available in literature through the total number of carbon (C) atoms present in the molecular formula and double-bond equivalence (DBE). Except the discussion of formation mechanism, the rest of the methodology and findings of this work is quite trivial and another repetition of multiple EEM spectroscopy based ambient aerosol characterization studies.

**Response**: We appreciate your valuable comments and suggestions, which can help improve the manuscript.

EEM fluorescence spectroscopy was initially employed to study chromophoric dissolved organic matter (CDOM) in terrestrial and oceanic systems before being used for atmospheric aerosol research. It has been suggested that organic chromophores differ across various environments, including aquatic systems and aerosols (H. Jiang et al., 2022; Wu et al., 2022). Studying the molecular compositions of fluorophores, specifically the fluorescent components decomposed by PARAFAC analysis, is crucial for improving our understanding of their structures, sources, and chemical properties in the atmosphere. Although the molecular compositions of PARAFAC components in aquatic systems have been well described (Kellerman et al., 2015; Stubbins et al., 2014), only a few studies have been conducted on these components in the atmosphere. Chen et al. (2016) investigated the chemical compositions of water-soluble PARAFAC components. However, they were unable to provide the corresponding molecular formulas using high-resolution aerosol mass spectrometers (HR-AMS). F. Jiang et al. (2022) and H. Jiang et al. (2022) provided respective molecular formulas for PARAFAC

components. However, their primary focus was on the methanol-extracted fractions and not water-extracts, which can significantly differ. To the best of our knowledge, only a recent study has provided the molecular formulas associated with water-soluble PARAFAC components in $PM_{2.5}$ collected in Guangzhou, a city in Southern China (He et al., 2023). However, they did not investigate the potential formation pathways of PARAFAC components and the molecular-level correlation between them and BrC. It is important to constrain the optical properties of BrC aerosols.

Additionally, our investigation discovered molecular formulas detected in the ESI+ mode that exhibit a significant correlation with PARAFAC components, providing a more comprehensive molecular characterization of them. The main objective of our study is to identify the molecular compositions and formation mechanisms of PARAFAC components in water-soluble organic carbon (WSOC), with a particular on identifying the relevant pathways, which, to our knowledge, have not been reported previously. Additionally, identification of diverse molecular compositions and formation pathways of commonly detected fluorescent components in the atmosphere will provide valuable information, particularly when utilizing EEM in combination with PARAFAC analysis to study atmospheric BrC. Thus, our work assists in comprehending the composition and fate of PARAFAC components and enhances the utilization of the EEM-PARAFAC method in characterizing atmospheric BrC. This is the novelty and significance of our work.

**References:**

He, T., Wu, Y., Wang, D., Cai, J., Song, J., Yu, Z., Zeng, X., and Peng, P. a.: Molecular compositions and optical properties of water-soluble brown carbon during the autumn and winter in Guangzhou, China, Atmos. Environ., 296, https://doi.org/10.1016/j.atmosenv.2022.119573, 2023.

Jiang, F., Song, J., Bauer, J., Gao, L., Vallon, M., Gebhardt, R., Leisner, T., Norra, S., and Saathoff, H.: Chromophores and chemical composition of brown carbon characterized at an urban kerbside by excitation–emission spectroscopy and mass spectrometry, Atmos. Chem. Phys., 22, 14971-14986, https://doi.org/10.5194/acp-22-14971-2022, 2022.

Jiang, H., Tang, J., Li, J., Zhao, S., Mo, Y., Tian, C., Zhang, X., Jiang, B., Liao, Y., Chen, Y., and Zhang, G.: Molecular Signatures and Sources of Fluorescent Components in Atmospheric Organic Matter in South China, Environ. Sci. Technol. Lett., 9, 913-920, https://doi.org/10.1021/acs.estlett.2c00629, 2022.

Kellerman, A. M., Kothawala, D. N., Dittmar, T., and Tranvik, L. J.: Persistence of dissolved organic matter in lakes related to its molecular characteristics, Nat. Geosci., 8, 454-U452, https://doi.org/10.1038/ngeo2440, 2015.

Stubbins, A., Lapierre, J. F., Berggren, M., Prairie, Y. T., Dittmar, T., and del Giorgio, P. A.: What's in an EEM? Molecular signatures associated with dissolved organic fluorescence in boreal Canada, Environ. Sci. Technol., 48, 10598-10606, https://doi.org/10.1021/es502086e, 2014.

Wu, G., Fu, P., Ram, K., Song, J., Chen, Q., Kawamura, K., Wan, X., Kang, S., Wang, X., Laskin, A., and Cong, Z.: Fluorescence characteristics of water-soluble organic carbon in atmospheric aerosol, Environ. Pollut., 268, 115906, https://doi.org/10.1016/j.envpol.2020.115906, 2021.

**Strength of this work:**

The authors analysed the FT-ICR MS data and came up with the most probable formation mechanisms from different molecular signals in addition to EEM spectroscopic studies.

Response: We appreciate the reviewer for providing valuable suggestions.

**Limitations of this work:**

*Lack of discussion on seasonal variation*: As seen in Figure 1, the study site is located at a very interesting geographical location with great seasonal variation in wind direction. In pre-monsoon and monsoon, the wind flow is directed from middle-east Asia and Arabian Sea, whereas the wind trajectories arise from North Pakistan and North-West India during post-monsoon and winter. This will result in very different chemical compositions of ambient particulate matter (PM) reaching the study site. For example, in the months of May-June, the aerosol composition will be close to marine aerosol composition, but in Oct-Dec there will be molecular signals of biomass burning emissions as previous studies have pointed out extensive crop burning and biomass burning for heat generation during winter in that part of the world. Which suggests that the wintertime aerosols will probably have higher S content and less oxygenated organics because the OH radical photochemistry is limited during winter and the atmospheric transformation is driven by NOx chemistry. The authors have reported all the data in a combined way, which does not give the readers the broader picture of the regional specific atmospheric chemistry of the study site. Postulating molecular formation pathways without considering the meteorological conditions can also lead to erroneous assumptions.

Response: Thank you for your suggestions. We chose this station as a typical region to study this topic due to its special geographical location. The station experiences great seasonal variation in wind direction, resulting in varying chemical compositions. This is evident from the differences in light absorption and fluorescence intensities across different seasons. Additionally, the molecular composition also differs across different seasons, a detail that was not shown in our original version. Our recent research has shown that continental-influenced WSOC has a higher composition of aromatic and highly oxidized compounds. Conversely, marine-influenced WSOC has a large availability of marine organic compounds that are saturated and have a lower degree of oxidation (Mo et al., 2022). This may highlight similarities in molecular characteristics between different air mass influences, compared to this study.

Because our primary objective is to obtain the molecular signatures and formation mechanisms of the fluorescent components decomposed by PARAFAC analysis, which is crucial for studying atmospheric BrC, we need subjective evaluations unless they are unambiguously marked as such. We appreciate your suggestions and as a result, we included the detailed molecular characteristics of each sample in different seasons in Tables S3 and S4 in the supplement for readers' reference.

Reference:

Mo, Y., Zhong, G., Li, J., Liu, X., Jiang, H., Tang, J., Jiang, B., Liao, Y., Cheng, Z., and Zhang, G.: The Sources, Molecular Compositions, and Light Absorption Properties of Water‐Soluble Organic Carbon in Marine Aerosols From South China Sea to the Eastern Indian Ocean, J. Geophys. Res.-Atmos., 127, https://doi.org/10.1029/2021jd036168, 2022.

*Lack of relevant references*: In continuation to the previous point, as the authors have not discussed the geographical context of this work, they have also failed to compare their findings with previous works carried out in similar locations. Although they have mentioned a few studies carried out in the Indo-Gangetic Plane (IGP) while reporting mass absorption efficiency (MAE) and Aerosol absorption exponent (AAE), this kind of comparisons have not been made for EEM spectroscopic studies. Previous EEM spectroscopic analyses of ambient aerosol in IGP and other parts of the world have found similar PARAFAC components (two HULIS and one protein like). These references from around the globe should be mentioned and compared with the findings of this work.

Response: Thank you for your suggestions. We regret our negligence in discussing the comparison with EEM spectroscopic studies. In order to improve the comprehension of the characteristics of fluorescent components in this geographic location, we have added a new discussion on the comparison in the revised manuscript. Please refer to the specified lines 290-307.

*Structure of the Results and Discussion Section*: The results and discussions section needs to be restructured. For the convenience of the readers, the discussion of

"Underlying implication of PARAFAC component to BrC absorption" should be done
under section 3.3 and the discussion on formation pathways should be under section
3.4. That way the flow of information will be more coherent.
Response: Thanks for the suggestions. In the revised manuscript, we have restructured
section 3.3 to discuss "Underlying implications of PARAFAC components on BrC
absorption" (lines 506-567) and section 3.4 to discuss "Potential formation mechanisms
of PARAFAC components" (lines 576-778).

The discussion on formation pathway of S-containing compounds should be more
condense and can also be moved to supplementary information (SI). The authors have
mentioned that S containing compounds have almost no effect on BrC chromophores.
They also reported that the two Humic-like PARAFAC components had very little S
containing compounds, mostly component C3 had the highest S containing compound.
Eventually it has also been shown that component C3 has the lowest overlap with the
BrC region in figure 5. Therefore, in terms of climate relevant BrC chromophores, this
pathway is not as important as CHO and CHON formation pathways.
Response: Thank you for your suggestion regarding the formation pathway of S-
containing compounds. According to previous studies, S-containing compounds have
little or no impact on BrC chromophores. Therefore, discussing S-containing
compounds may be confusing for readers. We have revised this section and moved it to
Text S7 in the revised supplement (lines 271-305).

The overall grammar and clarity of the current section 3.3 (proposed to be made section
3.4 after restructuring) is unsatisfactory. Too much statistics have been used in
sentences, which can instead be represented graphically. Sentences need to be written
with proper grammar, for example 407-408 is unclear and needs to be rewritten, so does
413. There are many occurrences like this, so the reviewer suggests a rewriting of this
whole section in a clear and concise manner. In the rewriting, the authors should also
elaborate and clarify how they reached conclusions made in 520-522.
**But most importantly in this section**, the authors have looked at some
molecular signals found through FT-ICR MS and compared those molecules with aged
byproducts of certain precursors reported in previous literature. And by doing that they
have tried to postulate these known reaction pathways to be present in their study
samples. These are good hypotheses, but there is no concrete way of establishing these
reaction pathways in the scope of this study. Therefore, it doesn't add up as novel new
information. If the authors can combine other analytical study with the collected filters
(if there is any remaining), for example [1]H NMR spectroscopy of WSOC, in which they can quantitatively compare the spectra of the precursor and aged molecules with their
sample spectra and confirm its presence, that would be a much stronger argument for
the formation pathways.

Response: We apologize for the low quality of the language in our manuscript. We spent
a long time revising the manuscript, which involved repeatedly adding and removing
sentences and paragraphs. This led to a significant decrease in readability. We have
worked on improving both the language and the readability of the document.
Additionally, we have invited fellow experts to make professional language editing for
the manuscript. We hope that the flow and language level have been significantly
improved.

Furthermore, the sentence in original lines 407-408 was modified as follows: "For
instance, Mo et al. (2018) demonstrated that 98% of CHON compounds found in $PM_{2.5}$
collected from Beijing have O/N $\geq$ 3. This finding indicates that the compound has at
least one nitro ($-NO_2$) or nitrooxy ($-ONO_2$) group in addition to other oxygen-
containing groups (i.e., $-OH$ and $-COOH$)." Please refer to lines 618-622 in the revised
manuscript.

The sentence in original lines 413 was modified as follows: "However, oxidized CHON
formulas assigned to C1 and C2 may be underestimated as they contain two or more N
atoms and do not require three folds of O atoms to form $-NO_2$ or $-ONO_2$ groups."
Please refer to lines 626-629 in the revised manuscript.

We draw the conclusion (original version in lines 520-522) based on the elemental
compositions of the overlapping molecules assigned to C1 and C2, which consist
mainly of CHON and CHO compounds. Additionally, the formation pathways of these
overlapping molecules were found to be a significant part of oxidation-derived
pathways, with CHON compounds primarily attributed to $N_2O_5$ oxidation and the CHO
compounds were highly detected in the SOA formed from limonene ozonolysis. For the
revised sections, please refer to lines 763-778 in the revised manuscript.

Additionally, we hope to explore the formation pathways of PARAFAC components by
comparing the assigned molecules to those reported in previous studies, despite the
potential for uncertainty. This approach may offer additional clarity on the possible
formation pathways of PARAFAC components. For example, when comparing the
CHO compounds assigned to PARAFAC components to the SOA formed from
limonene ozonolysis, our focus is on the presence of oxidized CHO compounds rather
than on the precursors. Also, when comparing to the CHO compounds resulting from
the photooxidation of naphthalene, biomass burning emissions, and aqueous-phase
reactions of phenols, we have obtained new insight into the high aromatic structures
and the predominance of C2 formulas in an oxidative form. As we concluded, the oxidation pathway appears to be the main pathway for the formation of C1 and C2. Indeed, the use of $^1$H NMR spectroscopy is important to characterizing WSOC and obtaining insight into their structural characteristics. However, there are no remaining samples for subsequent analysis. This study presents preliminary findings on the formation pathways of PARAFAC components, given their complexity. In the future, other analytical techniques, such as $^1$H NMR spectroscopy, should be combined to provide deeper insight into the formation pathways of fluorescent components.

**Minor corrections:**

Title: The title should be reconsidered. Instead of water soluble particulate matter, water soluble organic carbon is a preferable choice. The study location should be kept Karachi or mentioned a location in South East Asia.

Response: Thank you for the suggestions. The title has been revised as follows: "Molecular signatures and formation mechanisms of water-soluble chromophores in particulate matter from Karachi (Pakistan) in South Asia"

**Corrections in SI:**

Figure S2: The title of the figure mentions February 16, but the legend in the figure shows the trajectory starting from 17 February 2016.

Response: We apologize for the error. The figure has been revised by redrawing it due to the low resolution of the original version. We now present the updated figure below.

[Figure]

**Figure S2.** The 72 h back air-mass trajectories at Karachi from Pakistan for the selected samples via FT-ICR MS analysis on February 16, May 10, July 1, September 2, November 16, 2016, and January 20, 2017, correspond to different seasons. The trajectories with black dots represent the corresponding night, otherwise, it is the day. The air-mass trajectories were analyzed by the

HYSPLIT model. The map was created using Arcgis software, and the base map is from the National

Platform for Common Geospatial Information Services (www.webmap.cn).